# Direct production of low-oxygen-concentration titanium from molten titanium

Toru H. Okabe ●[1] ✉, Gen Kamimura ●[1], Takashi Ikeda ●[1] & Takanari Ouchi ●[1] ✉

Titanium (Ti) is an attractive material, abundant in nature and possessing superior mechanical and chemical properties. However, its widespread use is significantly hampered by the strong affinity between titanium and oxygen (O), resulting in elevated manufacturing costs during the refining, melting, and casting processes. The current work introduces a high-throughput technique that effectively reduces the oxygen content in molten titanium to a level suitable for structural material applications (1000 mass ppm, equivalent to 0.1 mass%). This technique aspires to streamline the mass production of titanium by seamlessly integrating the refining, melting, and casting processes. The developed method leverages the high affinity of rare-earth metals, such as yttrium (Y), for oxygen. This method utilizes the formation reaction of their oxyhalides (YOF) to directly remove oxygen from liquid titanium, resulting in titanium with a significantly reduced oxygen content of 200 mass ppm. This technique enables the direct conversion of titanium oxide feeds into low-oxygen titanium without requiring conversion into intermediate compounds. Additionally, this process offers a pathway for the upgrade recycling of high-oxygen-content titanium scrap directly into low-oxygen titanium. Consequently, this technology holds the potential to dramatically lower titanium production costs, thereby facilitating its more widespread utilization.

Titanium (Ti) and its alloys exhibit the highest specific strength and exceptional corrosion resistance among metallic materials. Despite its virtually limitless abundance as a resource, the widespread utilization of Ti products is constrained primarily due to the high affinity between Ti and oxygen (O), which poses challenges for the refining, melting, and casting processes[1]. Consequently, the production cost for Ti products remains high.

The current method employed for the industrial-scale production of metallic Ti (the Kroll process) involves the following reactions. High-quality or upgraded Ti oxide ore ($TiO_2$) is reacted with coke (C) and chlorine ($Cl_2$) gas at high temperatures of around 1273 K (1000 °C) to chlorinate Ti and convert it into oxygen-free titanium tetrachloride ($TiCl_4$). The purified $TiCl_4$ is then reduced using metallic magnesium (Mg) to produce metallic Ti. This chlorination method can

manufacture high-purity metallic Ti with an oxygen concentration of about 500 mass ppm (0.05 mass%). However, this process is characterized by limited productivity and high costs. It is energy-intensive, contributes significantly to carbon dioxide ($CO_2$) emissions, and imposes a substantial environmental burden. A further significant disadvantage of the conventional extraction route is the need to melt the Ti sponge produced from the Kroll process, which significantly increases costs and energy consumption. At present, however, the $TiCl_4$ route represents the only efficient method for removing oxygen from $TiO_2$ or high-oxygen-content Ti metal. There is thus an urgent need to develop a high-speed and efficient refining method to realize the mass production of low-cost Ti.

Other methods of removing oxygen from $TiO_2$ to produce metallic Ti involve the reduction of $TiO_2$ using aluminum (Al) or Mg,

[1]Institute of Industrial Science, The University of Tokyo, Tokyo, Japan. ✉e-mail: okabe@iis.u-tokyo.ac.jp; t-ouchi@iis.u-tokyo.ac.jp

both of which exhibit strong chemical affinities with oxygen. These processes are known as the aluminothermic reduction (ATR) and the magnesiothermic reduction (MTR)[2–7], represented by reaction Eq. (1) and (2), respectively:

$$TiO_2\,(s, \text{or in flux}) + 4/3\,Al\,(l) = Ti\,(s, l) + 2/3\,Al_2O_3\,(s, \text{or in flux}) \quad (1)$$

$$TiO_2\,(s, \text{or in flux}) + 2\,Mg\,(l, g) = Ti\,(s, l) + 2\,MgO\,(s, \text{or in flux}) \quad (2)$$

While Al and Mg can reduce $TiO_2$ to its metallic state, their ability to eliminate oxygen dissolved in the obtained metallic Ti−i.e., their deoxidation capability−is inadequate. As a result, Ti produced by ATR (ATR-Ti) or MTR (MTR-Ti) invariably retains substantial oxygen impurities, with a concentration of at least 10000 mass ppm (1 mass%). Since the oxygen concentration in Ti for structural applications is conventionally restricted to a maximum of 4000 mass ppm (0.4 mass%), ATR-Ti or MTR-Ti cannot be directly utilized as structural materials. Furthermore, ATR-Ti is heavily contaminated with Al, rendering it unsuitable for commercial applications in its current state[7].

The challenge of removing oxygen directly from Ti has been addressed with the development of techniques using metals with higher chemical affinities than Al and Mg as deoxidizing agents. In a previous study by the authors, a technique was developed to remove oxygen from Ti by reacting oxygen present on the surface of solid Ti with rare-earth metals, such as yttrium (Y), and their chlorides at about 1300 K (1027 °C)[8]. This technique, which reduces oxygen to a concentration of less than 100 mass ppm (0.01 mass%), is effective for deoxidizing Ti scrap, typically a few millimeters thick, which facilitates the recycling of Ti scrap. In this method, however, the diffusion of oxygen inside the solid Ti is the rate-controlling step. Removing oxygen from larger Ti samples, such as high-oxygen Ti ingots or ATR-Ti and MTR-Ti ingots, becomes time-intensive, making this approach impractical for the mass production of low-oxygen Ti suitable for structural applications.

A more favorable approach for quickly removing the oxygen present in large Ti ingots involves the elimination of oxygen from the liquid phase of Ti[9–14]. Molten Ti is commonly achieved with a high-frequency induction melting, plasma arc melting, or electron beam melting furnace. Notably, high-frequency induction melting can effectively agitate molten Ti, which enables the rapid production of homogeneous Ti or Ti alloys. However, molten Ti exhibits exceedingly high reactivity with oxygen, rendering oxygen removal challenging.

In past research, Kobayashi and Tsukihashi conducted deoxidation equilibrium experiments for molten Ti using Y with a cold crucible levitation melting furnace, a type of high-frequency induction melting furnace[9]. Their experiments demonstrated that adding Y can reduce the oxygen concentration in molten Ti to 2400 mass ppm (0.24 mass %). Similarly, Fukuzawa et al. investigated a method involving the addition of cerium (Ce), a rare-earth metal, or mischmetal, a mixture of various rare-earth metals, along with calcium fluoride ($CaF_2$), to liquid Ti[10]. This technique yielded the capability to lower the oxygen concentration in molten Ti to around 5000 mass ppm (0.5 mass%). Despite these efforts, the oxygen levels achieved were too high for structural material applications.

Furthermore, approaches to remove oxygen from molten Ti alloys with high Al content have been proposed. It has been demonstrated that oxygen content can be reduced in a molten Ti–46 mol%Al–8 mol%Nb alloy to 800 mass ppm (0.08 mass%) using Y and $CaF_2$[11]. Additionally, by melting a Ti–Al alloy containing over 40 mass% Al, oxygen concentrations can be reduced to 220 mass ppm (0.022 mass%)[12]. The thermodynamic stability of oxygen in Ti alloys containing a large amount of Al is lower than in pure Ti, allowing for the deoxidation of high-Al Ti alloys to low concentrations. However, these methods are inadequate for achieving reduced oxygen levels in pure Ti or Ti alloys with a low Al content, in which the thermodynamic stability of oxygen is high.

Given the context presented above, the current work proposes a method of effectively reducing impurity oxygen levels in molten Ti below 1000 mass ppm (0.1 mass%), thereby enabling the production of Ti with low oxygen concentrations. Leveraging the findings of the current work, as illustrated in Fig. 1a, can lead to the development of a new path for Ti refining that uses $TiO_2$ as a feedstock to produce low-oxygen-concentration Ti, eliminating the need for the conventional chloride refining process. Additionally, as shown in Fig. 1b, it becomes feasible to produce low-oxygen-concentration Ti directly from high-oxygen-concentration Ti scrap.

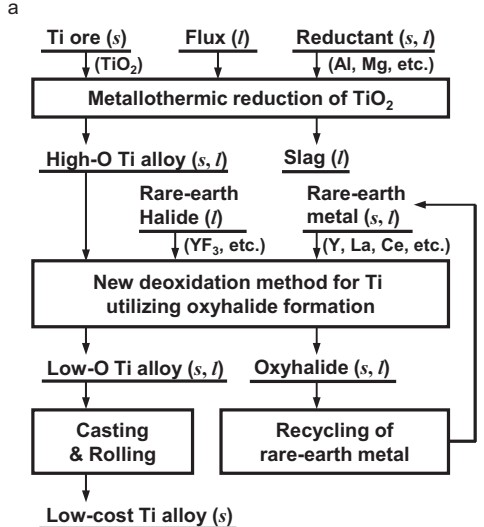

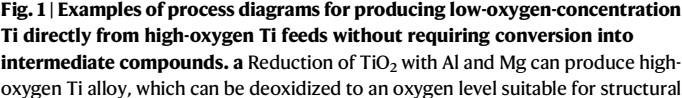

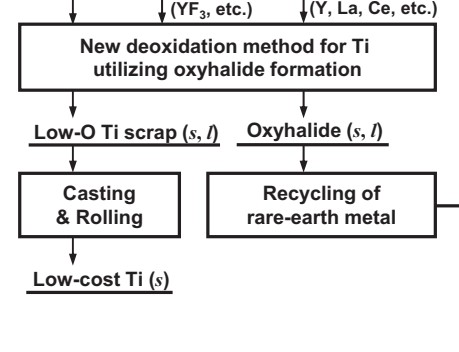

**Fig. 1 | Examples of process diagrams for producing low-oxygen-concentration Ti directly from high-oxygen Ti feeds without requiring conversion into intermediate compounds. a** Reduction of $TiO_2$ with Al and Mg can produce high-oxygen Ti alloy, which can be deoxidized to an oxygen level suitable for structural materials through the proposed method utilizing rare-earth metals and their oxyhalide formation. **b** High-oxygen Ti scrap from the fabrication of Ti ingots or used products can be directly converted to high-purity Ti with the proposed method. Phase of substances: (s) solid; (l) liquid.

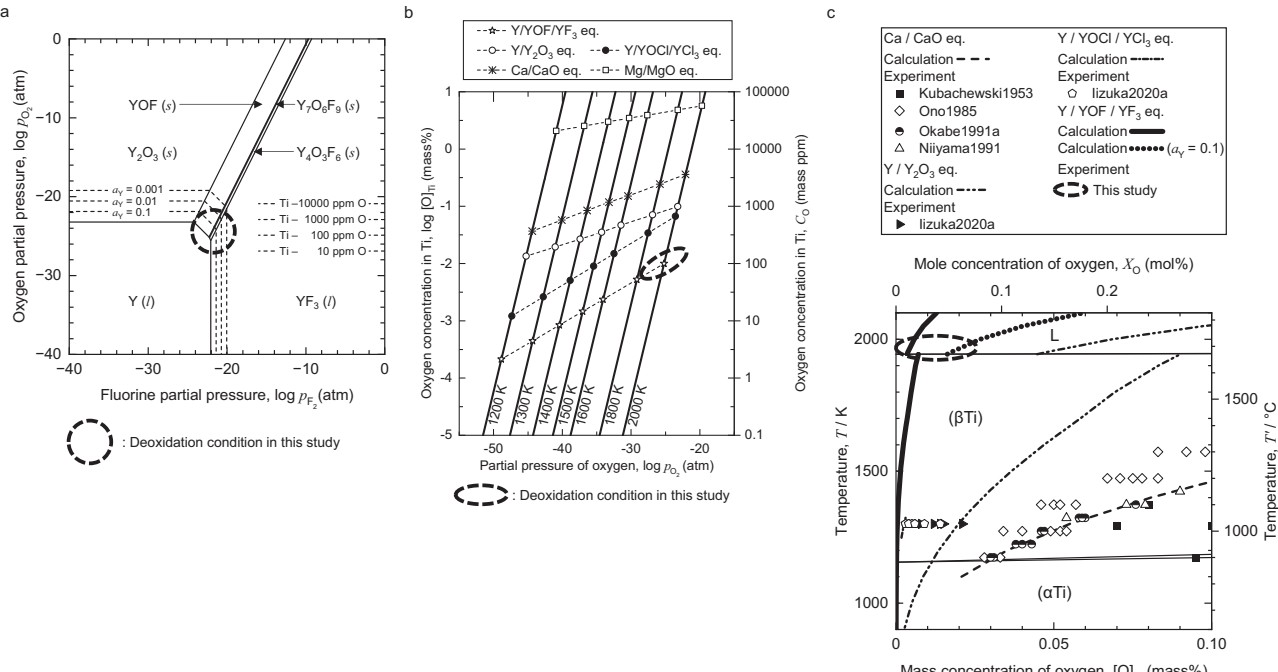

**Fig. 2 | Thermodynamic calculation of the deoxidation of liquid Ti using Y metal and YOF formation. a** Isothermal chemical potential diagram of the Y–O–F system at 2000 K (1727 °C). **b** Relationship between equilibrium oxygen concentrations in Ti and equilibrium partial pressures of oxygen under various temperatures. Equilibrium oxygen concentrations in Ti under various deoxidation reaction systems, such as Y/YOF/YF$_3$ eq., are also plotted. **c** Equilibrium oxygen concentrations in Ti under various deoxidation reaction systems, plotted in the phase diagram for the Ti–O binary system. The dashed-line areas in the figure indicate the deoxidation conditions investigated in the current work.

## Results

### Thermodynamic evaluation on deoxidation of liquid Ti

The feasibility of removing oxygen directly from Ti is assessed by thermodynamic evaluation[7,15–27]. The current work uses rare-earth metals as deoxidizing agents for molten Ti at temperatures over 1941 K (1668 °C). Additionally, rare-earth metal halides act as fluxes to effectively promote the deoxidation reaction. Unlike historical approaches utilizing fluxes, such as CaF$_2$, to dissolve the deoxidation products such as oxides (e.g., rare-earth metal oxides (REO$_x$)) in order to decrease their activity, the deoxidation reaction used in the current work involves rare-earth metals and their halides, as represented by reaction Eq. (3).

$$O\,(\text{in Ti}\,(l)) + 2/3\,RE\,(l) + 1/3\,REX_3\,(l) \rightarrow REOX\,(s,l) \quad (3)$$

Here, RE denotes rare-earth metals like Y and lanthanum (La), and X represents halogens like chlorine (Cl) and fluorine (F).

In this case, a distinct chemical pathway is pursued in which the formation of a rare-earth metal oxyhalide (REOX) as the deoxidation product can reduce the activity of the oxide. In the current work, Y is selected as the representative rare-earth metal (RE), and fluorine (F) is chosen as the halogen element (X). Among rare-earth metals, Y exhibits exceptionally high deoxidation properties. Fluoride compounds are often used as fluxes due to their high liquid-state stability, even at temperatures above the melting point of Ti. Here, the following reaction Eq. (4) will be discussed, which involves the deoxidation of Ti by the Y metal and yttrium oxyfluoride (YOF) formation.

$$O\,(\text{in Ti}\,(l)) + 2/3\,Y\,(l) + 1/3\,YF_3\,(l) \rightarrow YOF\,(s,l) \quad (4)$$

Thermodynamic data for compounds in the Y–O–F system at 2000 K (1727 °C) are compiled in Supplementary Table 1. Thermodynamic data for Y$_2$O$_3$ and YF$_3$ were derived from Barin's data

collections[24]. Baek and Jung's work shows that the liquid YF$_3$ phase can coexist with the yttrium oxyfluoride phases at temperatures as high as 1973 K (1700 °C), above the melting point of Ti[25]. Thermodynamic data for oxygen in liquid Ti at 2000 K (1727 °C) are also included in Supplementary Table 1. These data are scattered among the literatures and are characterized by significant uncertainty.

Figure 2a represents the chemical potential diagram for the Y–O–F system at 2000 K (1727 °C). The diagram also shows equilibrium oxygen concentrations in the molten Ti (10–10000 mass ppm (0.001–1 mass%)), corresponding to the oxygen partial pressure, $p_{O_2}$ (atm), on the vertical axis. Figure 2b depicts the thermodynamic relationship between equilibrium oxygen partial pressure and oxygen concentration in Ti, [O]$_{Ti}$ (mass%), in various deoxidation systems. Figure 2c shows equilibrium oxygen concentrations in Ti corresponding to various deoxidation systems in the Ti–O binary phase diagram. The dashed-line regions in these figures represent the equilibrium conditions of the chemical reactions employed in the current work. The activities of the deoxidant (Y), the flux (YF$_3$), and the reaction product (YOF)—with reference states corresponding to pure solid Y, pure liquid YF$_3$, and pure solid YOF, respectively—were set to unity in principle. The decrease in Y activity due to its dissolution into the Ti melt are considered in Fig. 2a, c. Given the substantial amount of YF$_3$ flux used in the experiments and the relatively low solubility of yttrium oxyfluoride in the liquid YF$_3$ phase at temperatures near the melting point of Ti[25], it is reasonable to assume unity for the activities of YF$_3$ and YOF.

Impurity oxygen is removed from the molten Ti, reducing oxygen concentration to a concentration of 100–1000 mass ppm (0.01–0.1 mass%) under equilibrium conditions ($p_{O_2} \approx 10^{-25}$ atm) where the metallic Y, YF$_3$ flux, and YOF deoxidation product coexist (Y/YOF/YF$_3$ equilibrium). Figure 2c also plots previous experimental studies[8,17–20]. While oxygen can be successfully reduced in solid Ti to a concentration level of 100 mass ppm (0.01 mass%) in the low-temperature region at

around 1300 K (1027 °C), there are no instances of deoxidizing molten Ti below 1000 mass ppm (0.1 mass%) at around 2000 K (1727 °C).

The outlined approach capitalizes on the deoxidizing capabilities of rare-earth metals and their oxyhalide formation, suggesting the potential for direct removal of oxygen from molten Ti to achieve a sufficiently low concentration based on the equilibrium state. However, the thermodynamic stability of rare-earth metal oxyhalides at high temperatures is largely uncharted, introducing the above calculations with substantial uncertainty. Therefore, deoxidation experiments for molten Ti were conducted to investigate the effectiveness of this deoxidation reaction system and its deoxidation limits.

### Deoxidation of liquid Ti utilizing Y metal and YOF formation

Figure 3 shows a cross-sectional schematic of the Ti melting apparatus used in the current work. Figure 4 shows photographs of the solidified

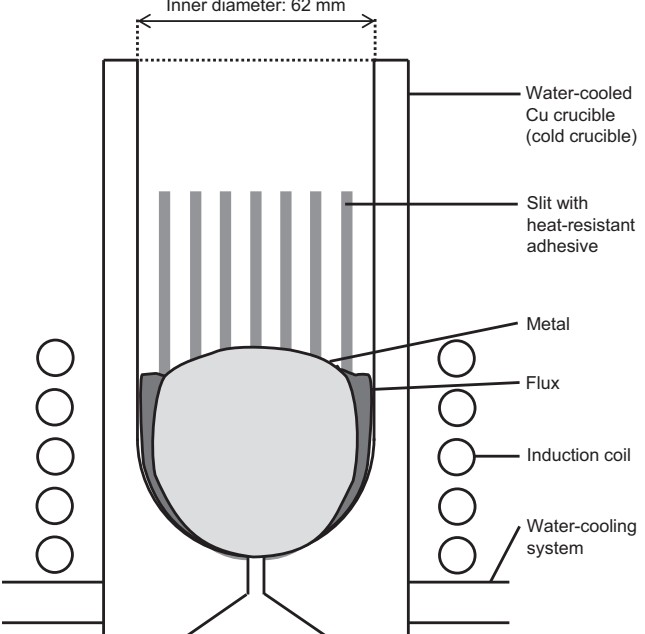

**Fig. 3 | Schematic diagram (cross-sectional view) of experimental apparatus for melting and deoxidizing Ti melt using a cold crucible high-frequency induction melting furnace.** Ti melt is levitated and agitated well by Lorenz force (pinch force) with induction heating, which can accelerate deoxidation and produce homogeneous Ti ingot. Flux can be removed to the outside of the ingot because the flux is not subjected to the Lorenz force. A heat-resistant adhesive is coated on slits in the cold crucible, facilitating metal recovery.

metal samples after the deoxidation experiment. The concentrations of oxygen and nitrogen (N) in the metal samples after the experiments are shown in Table 1, together with the experimental conditions. Figure 4d shows the Ti pieces for analysis in Exp. #1, which were sampled by cutting the deoxidized Ti ingot.

In Exp. #1, a metal sample with an extremely low oxygen concentration of 110–230 mass ppm (0.011–0.023 mass%) was obtained. Details of the analytical results for this Ti sample are shown in Supplementary Fig. 2 and Supplementary Table 2. The oxygen concentration decreased to 200 mass ppm (0.02 mass%) in the entire sample area, which agreed well with thermodynamic predictions. This result suggests that deoxidation uniformly proceeded throughout the entire sample. The nitrogen concentration in this metal sample was 120–150 mass ppm (0.012–0.015 mass%), and the Y concentration was around 2 mass%. Generally, the production of Ti with an oxygen concentration of less than 200 mass ppm (0.02 mass%) using the conventional industrial method via the chloride route poses significant challenges. The current work demonstrates for the first time the effectiveness of a novel refining method that removes oxygen directly from liquid Ti, reducing the oxygen concentration to extremely low levels.

In Exp. #2, a metal sample with a low oxygen concentration of about 600 mass ppm (0.06 mass%) was obtained. Details of the analytical results for the Ti sample in Exp. #2 are shown in Supplementary Figs. 3 and Supplementary Tables 3. The Y concentration in this metal sample was less than 1 mass%. This result demonstrates that oxygen could be reduced to a low concentration of around 1000 mass ppm (0.1 mass%) while minimizing the incorporation of Y into the deoxidized Ti melt.

Under the conditions of Exps. #6, #7, and #8, even when $TiO_2$ was intentionally added to attain the initial oxygen concentration of 10000 mass ppm (1 mass%) in Ti, the oxygen content was reduced to about 1000 mass ppm (0.1 mass%) in molten Ti. Details of the analytical results for the Ti sample in Exp. #8 are shown in Supplementary Figs. 4 and Supplementary Table 4.

In the melting experiments (Exps. #9, #10, and #11), it was also confirmed that Ti with a low oxygen concentration could be obtained when using La or Ce as the deoxidizing agent.

Conversely, in the control experiment (Exp. #13), in the absence of a rare-earth metal halide in the system, a metal sample with an oxygen concentration of less than 1000 mass ppm (0.1 mass%) was not obtained.

Figure 5 shows the X-ray diffraction (XRD) measurement results for the fluxes obtained after the melting experiments. As illustrated in the figure, it was found that yttrium oxyfluorides were present in the fluxes after the experiments, with no oxides (e.g., $Y_2O_3$) detected. Supplementary Fig. 6 and Supplementary Table 5 present the scanning

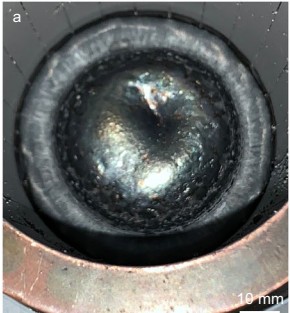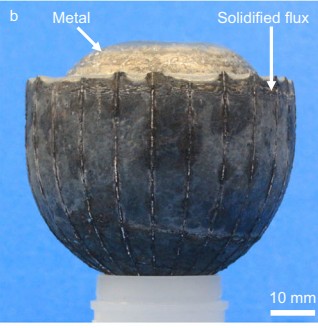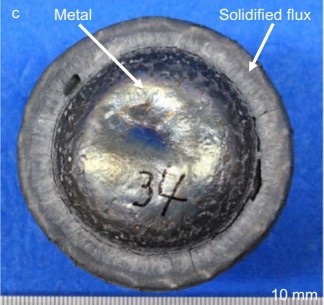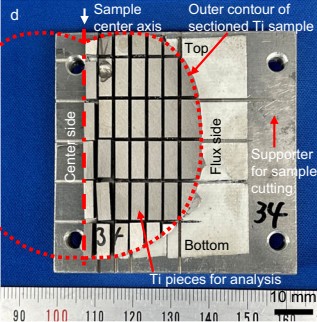

**Fig. 4 | Photographs of a Ti sample after melting and deoxidation using a cold crucible induction melting furnace (Exp. #1 in Table 1). a** The sample in the cold crucible after melting; **b** the side and **c** top views; and **d** sectioned Ti sample pieces for analysis. The metal sample was separated from the flux phase after induction melting.

**Table 1 | Experimental conditions of deoxidation of Ti melt, and analytical results of oxygen and nitrogen in Ti metal obtained after deoxidation**

| Exp.# | Materials charged into induction melting, $w_i$ / g | | | | | | | | | Analytical results of Ti samples after deoxidation | |
|---|---|---|---|---|---|---|---|---|---|---|---|
| | Ti | Y | La | Ce | $YF_3$ | $LaF_3$ | $CeF_3$ | $CaF_2$ | $TiO_2$ | Oxygen concentration, $C_O$ (mass ppm) | Nitrogen concentration, $C_N$ (mass ppm) |
| 1 | 350 | 12 | - | - | 110 | - | - | - | - | 170, 220, 110, 120, 140, 230, 110, 120, 190, 120 | 140, 140, 130, 120, 130, 150, 140, 120, 140, 150 |
| 2 | 350 | 1.6 | - | - | 110 | - | - | - | - | 600, 580, 620, 580, 600 | 100, 100, 100, 70, 110 |
| 3 | 350 | 65 | - | - | 110 | - | - | - | - | 870, 1300, 920, 1100 | 300, 300, 340, 310 |
| 4 | 350 | 65 | - | - | 110 | - | - | - | - | 680, 590, 760, 1100 | 180, 200, 210, 220 |
| 5 | 350 | 65 | - | - | 110 | - | - | - | - | 750, 920, 850, 730, 730 | 180, 180, 190, 180, 170 |
| 6 | 350 | 65 | - | - | 110 | - | - | - | 9.0 | 930, 850, 940, 700, 820, 820, 870, 790, 820 | 210, 220, 220, 180, 210, 180, 210, 210, 210 |
| 7 | 350 | 65 | - | - | 110 | - | - | - | 9.0 | 1400, 980, 1100, 1100, 1000, 1200 | 130, 160, 160, 140, 140, 170 |
| 8 | 350 | 65 | - | - | 110 | - | - | - | 9.0 | 1100, 860, 860, 780 | 240, 220, 230, 230 |
| 9 | 350 | - | 19 | - | - | 140 | - | - | - | 730, 750, 740, 740, 710 | 80, 90, 80, 90, 90 |
| 10 | 350 | - | 100 | - | - | 150 | - | - | 9.0 | 510, 500, 500, 980, 510 | 60, 50, 50, 90, 60 |
| 11 | 350 | - | - | 100 | - | - | 150 | - | 9.0 | 550, 570, 570, 570 | 150, 150, 130, 140 |
| 12 | 350 | 65 | - | - | - | - | - | 57 | - | 930, 870, 1400, 1300 | 150, 140, 160, 150 |
| 13 | 350 | 65 | - | - | - | - | - | - | - | 1400, 1400, 1200, 1900, 1100 | 170, 170, 150, 160, 130 |

Initial charged Ti samples: ~1000 mass ppm O, ~30 mass ppm N.
Melting conditions: Power input ~100 kW in Ar atmosphere (500 Torr) for 10–20 min.

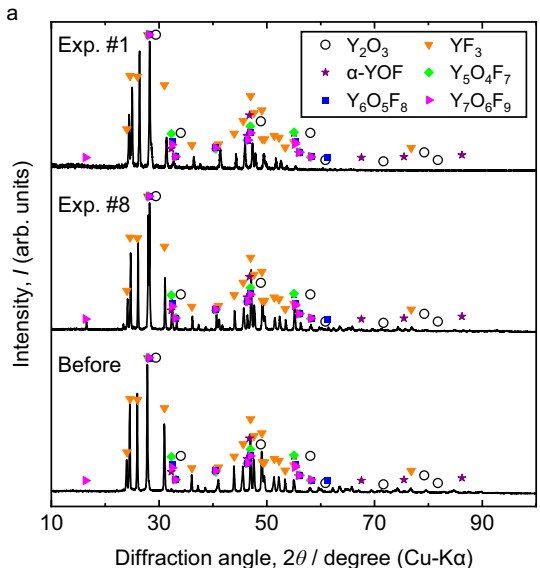
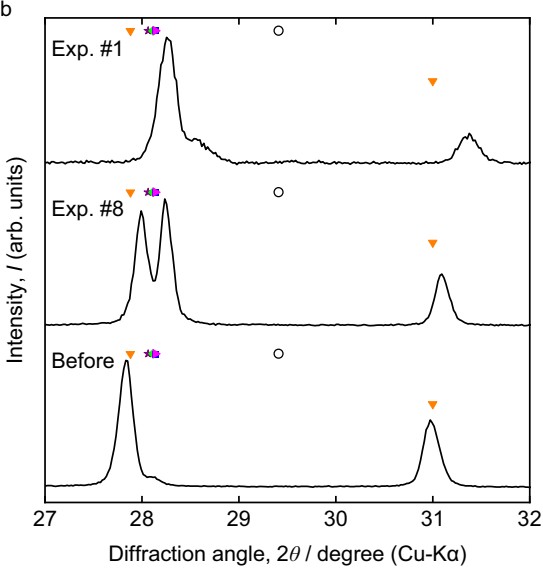

**Fig. 5 | XRD patterns of $YF_3$ fluxes before and after deoxidation.** The intensities are shown in (**a**) wide (10–100 degrees) and (**b**) narrow (27–32 degrees) diffraction angle scales. Various yttrium oxyfluoride phases were detected in the fluxes after the deoxidation experiments, whereas no yttrium or titanium oxide phases were observed, which shows that the Y/YOF/$YF_3$ equilibrium was established in the current systems.

electron microscopy (SEM) images and energy-dispersive X-ray spectroscopy (EDS) analysis for the fluxes from Exps. #1 and #8, where no $Y_2O_3$ phase was observed. From these results, it is inferred that, under the current experimental conditions, oxygen removal in Ti proceeded via the reactions indicated by Eqs. (3) and (4).

### Anticipated developments
Current technologies do not allow for the large-scale and high-speed manufacture of Ti with a low oxygen concentration directly from oxides and scrap without employing a chloride refining method. In contrast, the current approach enables direct production of Ti as low as 200 mass ppm O (0.02 mass% O) from $TiO_2$ in a short processing time, as shown in Fig. 1a.

Furthermore, as depicted in Fig. 1b, employing this method enables the production of Ti with a low oxygen concentration from Ti scrap that contains impurity oxygen. This facilitates Ti scrap reutilization, offering the chance to upgrade and recycle Ti scrap generated during both the fabrication of Ti ingots and Ti scrap from used products. This approach has the potential to significantly reduce $CO_2$ emissions associated with Ti refining.

In recent years, the increased production of magnet materials such as neodymium (Nd), dysprosium (Dy), and terbium (Tb) has led to an oversupply of Y, La, and Ce, which are produced simultaneously as by-products[28–33]. Consequently, processes using these abundant rare earth metals are becoming economically viable and resource-efficient. Furthermore, it is also technically possible to regenerate Y metal and $YF_3$ from YOF using methods such as molten salt electrolysis.

Figure 6 illustrates the integrated process proposed in the current work for Ti refining, melting, and casting, as well as the process for separating Ti and oxygen from $TiO_2$ without using a chloride refining

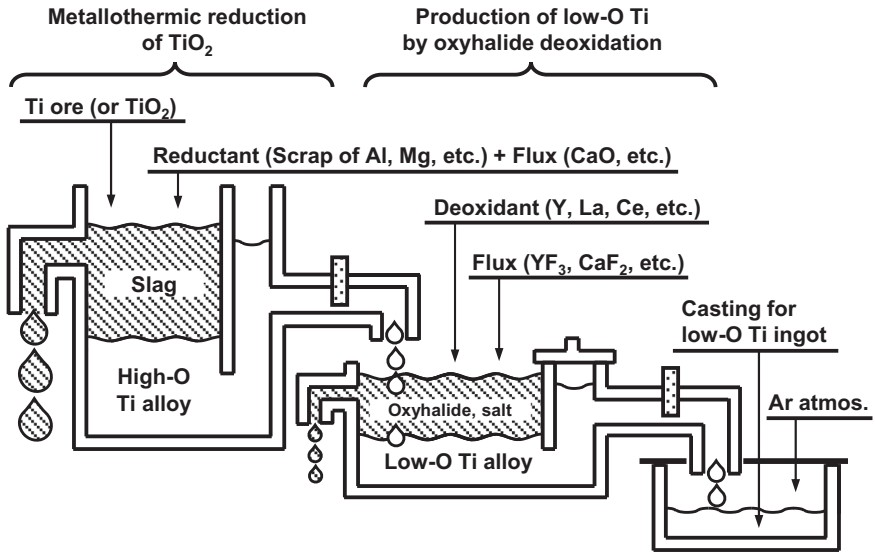

**Fig. 6 | Conceptual diagram of a new smelting method for producing Ti or Ti alloy from TiO₂.** The proposed deoxidation technique is designed to establish the mass production of titanium through the seamless integration of the refining, melting, and casting processes.

method. In the first step of this process, Al scrap or Mg scrap can be used as reducing agents for $TiO_2$. This approach enables the manufacture of high-oxygen-concentration Ti alloys without $CO_2$ emissions. Subsequent removal of oxygen from these high-oxygen alloys results in the production of extremely low-oxygen-concentration Ti. The deoxidation method for molten Ti demonstrated in the current work is applicable in various types of melting furnaces. For melting large quantities of Ti on an industrial scale, a larger-scale furnace, such as a plasma arc furnace, would be better suited than the induction skull furnace used in the current work.

If high-purity metallic Ti can be directly produced utilizing abundant Ti ores and Al scrap, the potential for cost-effective mass production of Ti arises. This would open the door to the widespread utilization of Ti products. The findings of the current work can develop the process technology to transform Ti into a widely used base metal, thereby facilitating its broad adoption across various sectors of society.

## Discussion

The current work has presented a technique to reduce oxygen directly from molten Ti below 1000 mass ppm (0.1 mass%). Specifically, the technique can manufacture Ti with a low oxygen concentration of (0.02 mass%) by heating, melting, and reacting Ti with rare-earth metals and their halides in a high-frequency induction furnace. The utilization of the formation reaction of rare-earth metal oxyhalides plays a central role in this technique. This technique holds the potential to establish a refining process for manufacturing high-purity Ti directly from $TiO_2$, eliminating the need for chloride refining. Furthermore, this technique also enables the upgrade recycling of high-oxygen-concentration Ti scrap. In the future, this method has the potential to become a new technique for producing Ti, thereby offering an alternative to the Kroll process, the current standard for Ti mass production.

## Methods
### Deoxidation experiments
**Experimental setup.** The melting apparatus is a cold crucible induction melting furnace, a type of high-frequency induction heater, with a maximum output of 180 kW. It consists of a water-cooled copper (Cu) crucible (internal diameter 62 mm, depth 130 mm) with slits to facilitate the permeation of magnetic fields, an induction coil encircling the

crucible, and a water-cooling system for the crucible. The induction coil is connected to a high-frequency power source, which was adjusted to maintain the Ti in a molten state.

**Sample preparation.** For typical deoxidation experimental conditions, 350 g of Ti rod and tube (initial oxygen concentration ~1000 mass ppm (0.1 mass%)) were placed inside the water-cooled Cu crucible, together with 2–65 g of Y chunk and 110 g of $YF_3$ powder. In some experiments (Exps. #6, #7, #8, #10, and #11), 9 g of $TiO_2$ powder was also added to attain an initial oxygen concentration of 10000 mass ppm (1 mass%) in the molten Ti.

**Heating and melting procedure.** The metal samples were heated and melted under 500 Torr in an argon (Ar) atmosphere. After visually confirming that the entire sample had melted, this molten state was maintained for 10–20 min (the melting condition: an output of 100 kW, a frequency of 39 kHz). The cold crucible induction melting furnace (induction skull melting furnace) used in the experiment is thought to have heated only to about 50 K above the melting point of the material. This is due to the large contact area between the water-cooled Cu crucible and the molten Ti (350 g) and the presence of solid material at the interface. Previous measurements indicate that the furnace does not heat the metal to more than 100 K above its melting point. This is because the stirring of the melt and the resulting radiative heat loss are substantial relative to the power input. When melting, it was visually confirmed that the molten Ti was strongly agitated by electromagnetic forces. Subsequently, the furnace output was reduced, and the samples were air-cooled and removed from the furnace.

**Post-experiment analysis.** Figure 4a shows photographs of the solidified metal samples inside the water-cooled Cu crucible after the experiment. Figure 4b, c depict the appearance of the samples after being taken out from the water-cooled Cu crucible. The samples were sliced and cut into small pieces (see Fig. 4d), and the concentrations of oxygen and nitrogen in the samples were quantified by the inert gas fusion method. The concentration of rare-earth elements in the samples was quantified by inductively coupled plasma atomic emission spectroscopy (ICP-AES), X-ray fluorescence spectroscopy (XRF), and SEM-EDS. SEM-EDS analysis and XRD measurement were used to identify the existing phases in the flux obtained after the experiment.

## Data availability
The authors declare that the data supporting the findings of this study are available within the paper, its supplementary information files.

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

## Acknowledgements
We express our gratitude to Mr. Koichi Hirota of Shin-Etsu Chemical Co., Ltd. for providing the rare-earth metal samples. Additionally, we thank Mr. Fumiaki Kudo of Kobelco Research Institute, Inc. for assisting with the melting experiments, as well as Mr. Yoichi Warashina of Toho Technical Service Co., Ltd., and Dr. Norio Yuki of Toho Titanium Co., Ltd. for their cooperation in providing the Ti and Ti alloy samples. We deeply appreciate the experimental assistance and advice for the manuscript preparation by Mr. Tomoki Yamazaki, a graduate student in the Department of Materials Engineering in the Graduate School of Engineering at the University of Tokyo. This research was supported by a Grant-in-Aid for Scientific Research (S) (Grant Number: JP19H05623) from the Japan Society for the Promotion of Science.

## Author contributions
T.H.O. conceived the basic invention, completed the experimental concept, and gathered funds and materials. G.K., T.I., and T.O. conducted the melting experiments and the analysis, evaluation, and interpretation and also prepared the figures. T.H.O., G.K., and T.O. prepared the manuscript.

## Competing interests
The authors have filed a patent covering the process described in this manuscript. PCT International Application. Patent applicant: The Foundation for the Promotion of Industrial Science (institution). Name of inventors: Toru H. Okabe, Gen Kamimura, Takashi Ikeda, and Takanari Ouchi. Application number: PCT/JP2024/015012. Status of application: under review. Specific aspect of manuscript covered in patent application: Method for producing titanium or titanium alloys
Japanese Patent Application. Patent applicant: The Foundation for the Promotion of Industrial Science (institution). Name of inventors: Toru H. Okabe, Gen Kamimura, Takashi Ikeda, and Takanari Ouchi. Application

number: 2023-117129 Status of application: under review. Specific aspect of manuscript covered in patent application: Method for producing titanium or titanium alloys.
