## [Peer Review File · Nature Communications]

Direct Production of Low-Oxygen-Concentration Titanium from Molten TitaniumREVIEWER COMMENTS

Reviewer #1 (Remarks to the Author):

The manuscript presents a new method to remove oxygen impurities from molten titanium metal using yttrium and yttrium oxyfluoride, achieving deficient oxygen concentrations of 200 ppm. The approach enables direct refining of titanium from titanium oxide feedstocks without chloride processing. It also facilitates the recycling of titanium scrap by reducing high oxygen levels. To be ready for publication, the manuscript will benefit from addressing the following comments:

1. The thermodynamic calculations provide an important basis for the approach but lack some key details. What sources were used for the rare earth oxyfluoride properties at high temperatures? How reliable are extrapolations to 2000K? What assumptions went into determining activities/ partial pressures? This affects the confidence in the predictions.
2. For the thermodynamic calculations, what data sources were used for properties of the Y-O-F system at elevated temperatures? Were these extrapolated? This affects the uncertainty.
3. The experimental results are promising, but measurements of additional samples would make the conclusions more robust. With only 1-2 samples for most conditions, variability between samples is unclear.
4. Agitation of the melt is mentioned but not characterised. Mixing likely plays a role in deoxidation rate and uniformity. Quantifying mixing time/velocity would aid understanding.
5. In Table 1, please add units for the materials charged (e.g., g).
6. How was mixing/agitation of the molten Ti characterised? Is it well mixed to enable uniform oxygen removal?
7. The integrated refining process in Fig. 6 is interesting but remains conceptual. Significant engineering development is needed to implement this commercially. The challenges should be discussed.
8. The recycling of rare earth metals is mentioned but not elaborated on. Details would strengthen the economic/sustainability case.
9. The Conclusions section focuses narrowly on the novel deoxidation approach but could be broadened to the larger implications for Ti production.
10. How was YOF stability characterised at high temperature? What indications are there that it persists to 2000K?
11. Were the samples crafted into test specimens to measure mechanical properties and validate usability?
12. How rapidly does deoxidation occur? Is there an optimum mixing speed?
13. What level of rare earth contamination in the Ti is acceptable for various applications?
14. What prior attempts have been made at direct oxide-to-metal smelting processes? Why did they fail or not advance?
15. Define acronyms first mention (e.g. ATR)
16. Avoid directional language like "this study" - replace it with "the current work"
17. Reduce the use of "we"
18. Tighten wording in places for brevity and clarity
19. Check references to match journal style
20. Abstract: Change "Endowed with superior mechanical and chemical properties" to "Possessing superior mechanical and chemical properties."
21. Line 53: Change "invariable retains" to "invariably retains"
22. Line 81: Change "imbuing" to "introducing"
23. Line 347: Add a space between "J58719A1" and the comma
24. Add Oxford comma in a few places with lists of 3 items

In summary, while the deoxidation approach shows promise, the paper would benefit from strengthening the thermodynamic and experimental sections, better contextualising the proposed process, and expanding the conclusions.

Reviewer #2 (Remarks to the Author):

see attachment

Reviewer #3 (Remarks to the Author):

This paper presents a method and experimental examples of the deoxidation of molten titanium.

Merits:

1. Deoxidation is a very challenging problem for Ti. The low oxygen content, <200 ppm, presented in the article is remarkable.
2. The process presented could be used for the production of primary Ti as well as the deoxidation of scrap Ti.

Detractors:

1. The novelty of the presented work is questionable. The authors' group has been very active in this area. I easily found, using Google search, over ten publications by the group on the general subject of using rare earth elements, specifically Y, to deoxidate Ti. Some are for deoxidate solid Ti, some molten Ti. I tried to see if the use of flux to form Y oxychloride is novel, but, the following publication seems to have significant similarity.

TY - JOUR

AU - Okabe, Toru

AU - Taninouchi, Yu-ki

AU - Zheng, Chenyi

PY - 2018/08/31

SP -

T1 - Thermodynamic Analysis of Deoxidation of Titanium Through the Formation of Rare-Earth Oxyfluorides

VL - 49

DO - 10.1007/s11663-018-1386-5

JO - Metallurgical and Materials Transactions B

2. The presentation of the article is a little unconventional. The authors presented examples number 1, 2, 3, etc. It reads like a patent application. The style makes it difficult to grasp the method's principles beyond the data report.
3. Another very minor point. XRD cannot be used to determine if residual oxides of other elements are present. It is not sensitive enough.

Review of

T. H. Okabe et al.: „Direct Production of Low-Oxygen-Concentration Titanium from Molten Titanium”

General comment on authors:

T. H. Okabe has published many important articles on alternative methods for titanium metal extraction during the last almost twenty years, e.g. on calciothermic and magnesiothermic reduction, usage of halides and electrochemical methods. In 2019 and 2020, respectively, he authored two review papers in this field and a book chapter (reference 1 in current work). Recently, he published several articles on the usage of rare earths in combination with halide fluxes, as kind of predecessors to the current work.

The co-authors have published several articles on extractive metallurgy of titanium as well. Therefore, it is safe to say that the authors of the current work have got a high expertise in this field.

General comments on the approach described in current work:

The *key aspect* of the approach is the deoxidation of titanium metal by rare earth metals (e.g. yttrium Y) and a halide flux (e.g. yttrium fluoride YF₃) with the formation of oxyhalides (e. g. YO₂) instead of corresponding oxides, thus achieving significantly lower oxygen-contents.

The *novelty* of this approach is unquestionable.

The *soundness* of this approach is made plausible by suitable thermodynamic calculations and lab-scale experiments.

The paper is in principle suitable for publication in Nature Communications.

General criticisms/suggestions:

- Contamination of the deoxidized titanium by yttrium is significant, as evident from the current work and several preceding publications. This issue is not sufficiently addressed in the current work. Experimentally obtained Y-contents are only roughly stated in the text, but instead they should be precisely given in Table 1 together with the contents of oxygen and nitrogen. In this way, the varying oxygen-contents become also more understandable (solubility product of Y and O, according to equation 3 and 4). According to section “post-experiment analysis”, Y and the other rare earth elements have been analyzed by ICP-AES and XRF, thus their exact contents should be given in Table 1. In addition, a general remark could be appropriate that the consequences of significant contents of the rare earths in titanium on its properties in the final application are not yet sufficiently investigated to date.
- Temperature control in the experiments is not clear. Induction melting, in particular with a cold crucible, generally is challenging regarding temperature control of the melt. The temperature in turn is crucial to the thermodynamic equilibrium of the deoxidation (e. g. equation 4). It should be stated how the temperature was measured (radiation pyrometer, thermocouple etc.) and which values or range were obtained.

- Application of cold crucible (“scull”) induction melting possibly limits the potential for industrial scale-up and is very energy intensive. However, it is clear from the current state of the art and science that suitable refractory crucibles have not been found yet for high-Ti melts despite decades of intensive research and development. Y_2O_3 has been applied for Ti-melts in prior publications, but would not be suitable in the current approach due to the need of saturation of the flux with Y_2O_3 in that case. So, it remains open what refractories could be used in the process sketched in Figure 6. Nevertheless, compared to conventional melting processes for titanium (vacuum arc remelting, electron melting and plasma arc melting), cold crucible induction melting could still be advantageous.

There are the following comments and suggestions:

- Equations (3): One should assume that the RE is dissolved in the Ti and that REX_3 and REOX form a homogeneous mixture phase. At 1700°C Ti and Y are completely miscible. Then, the equation reads:

As a consequence, the reduced activities of the corresponding component have to be considered in the thermodynamic calculations.

- Figure 2b: E.g. according to the reaction

the mole fraction of x_O in Ti is given by

$$x_O = \frac{1}{\gamma_O} \left(\frac{K(T)a_{Y_2O_3}}{a_Y^2} \right)^{1/3}$$

with $K(T)$ equilibrium constant and γ_O the activity coefficient of O in Ti (which in principle also depends on x_Y). The equation can also be written as

$$P_{O_2} = a_O = \left(\frac{K(T)a_{Y_2O_3}}{a_Y^2} \right)^{1/3}$$

(O_2 reference state for oxygen). So, the position of the hollow circles in Figure 2b depend on a_Y (and thereby on x_Y) and on $a_{Y_2O_3}$. Of course, the same is also the case for the other symbols. It is o.k. to make a reasonable choice for the activities (e.g. $a=1$) but it should be explained more clearly which one and why.

Minor comments and suggestions:

- It's good practice to mention the reference state when using activities like in Figure 2a.
- Line 45 etc.: A further significant disadvantage of the conventional extraction route is the need for remelting the titanium sponge from Kroll process, thus increasing costs and energy demand significantly.
- Line 48: Expression "...both of which exhibit higher chemical affinities with oxygen than Ti" is imprecise and not quite right. As also evident from Figs. 2b and 2c, due to the significant oxygen solubility of titanium, the oxygen potential P_{O_2} required for reduction/deoxidation is delicately determined by the pursued oxygen content.
- Line 60 etc.: Regarding aluminothermy, the significant contamination of titanium with aluminium could be mentioned, according to the solubility product in the system Ti-Al-O. Furthermore, the seminal work of Maeda et al. (reference 22) could also be cited here.
- Line 60 etc.: Why is calciothermy not mentioned beside aluminothermy and magnesiothermy? Due to costs for calcium?
- Line 183: "... while suppressing the incorporation of Y into the deoxidized Ti melt". It would be better to write "... while minimising the incorporation of Y into the deoxidized Ti melt". A "suppressing" of Y-pick up is thermodynamically impossible.
- Line 200 (Fig. 3): Although the apparatus dimensions are later given in the section "experimental setup", why not draw the internal diameter of the crucible or a scale right in the sketch?
- Line 210 (Fig. 4c): Is the outer "ring" of the sample the solidified flux? (could be mentioned in the corresponding caption)
- Figure 4d: The Figure is difficult to understand.
- Line 218 (Table 1): The last column ("Experimental code") contains no information relevant to the public, unless it refers to supplementary data additionally provided. It would be very useful to give also the Y-, La-, Ce-content in the final material.
- Line 260 (Fig. 6): Designation "Flux (CaF₂, etc.)" could be extended by "YF₃", according to the approach and findings of the current work.
What are the designated refractory materials?
- Line 296: Pressure of argon atmosphere (e.g. "ambient")?
- Line 356: "...seoxidation" (typo).

Answers for comments by Reviewers

REVIEWER COMMENTS

Reviewer #1 (Remarks to the Author):

The manuscript presents a new method to remove oxygen impurities from molten titanium metal using yttrium and yttrium oxyfluoride, achieving deficient oxygen concentrations of 200 ppm. The approach enables direct refining of titanium from titanium oxide feedstocks without chloride processing. It also facilitates the recycling of titanium scrap by reducing high oxygen levels.

To be ready for publication, the manuscript will benefit from addressing the following comments:

1. The thermodynamic calculations provide an important basis for the approach but lack some key details. What sources were used for the rare earth oxyfluoride properties at high temperatures? How reliable are extrapolations to 2000K? What assumptions went into determining activities/ partial pressures? This affects the confidence in the predictions.

Thermodynamic calculations for Y_2O_3 and YF_3 in the current work relies on data from Barin's collection (Ref. 24) [Barin_1995]. However, such general data collections lack comprehensive data for yttrium oxyfluorides like YOF. This gap was filled by using data on the thermodynamic properties of YOF derived from Levitskii and Balak's pioneering work, based on electromotive force measurements using solid electrolytes at temperatures near 1273 K (1000 °C) (Ref. 26) [Levitskii_1982]. Furthermore, Baek and Jung have experimentally measured the Y_2O_3 -YF pseudobinary phase diagram and its thermodynamic properties for temperatures up to 1973 K (1700 °C) (Ref. 25) [Baek_2022]. Their findings indicate that the YF_3 liquid phase remains in equilibrium with yttrium oxyfluoride phases even above Ti's melting point. These high-temperature data, extrapolated from Levitskii and Balak's lower-temperature findings, have been integrated into the current work. However, data on YOF thermodynamics at high temperatures are scarce and unreliable. They can, moreover, be marred by significant experimental errors, particularly when extrapolated to higher temperatures. Consequently, thermodynamic predictions in the current work are acknowledged as somewhat approximate.

In Previous research, the authors thermodynamically analyzed the deoxidation reactions of Ti at temperatures around 1300 K (1027 °C), particularly focusing on the formation of rare-earth metal oxychlorides in chloride systems [Okabe_2018_a] and fluoride systems [Okabe_2018_b]. Subsequently, deoxidation equilibrium experiments were conducted based on this analysis. The results showed that there was no significant divergence between the predicted values calculated from thermodynamic data and the experimental results. (Y system: [Iizuka_2020], La system: [Tanaka_2020], Ce system: [Kamimura_2022], Ho system: [Kong_2021]) The current study also demonstrates a close correspondence between thermodynamic predictions and experimental values, suggesting that a relatively reliable thermodynamic analysis has been conducted.

The literatures on the deoxidation of Ti using rare-earth metals and their halides at temperatures around 1300 K are as follows:

Thermodynamic considerations:

RE-chloride [Okabe_2018_a]

Okabe, T. H., Zheng, C. & Taninouchi, Y. Thermodynamic Considerations of Direct Oxygen Removal from Titanium by Utilizing the Deoxidation Capability of Rare Earth Metals. *Metall. Mater. Trans. B* **49**, 1056–1066 (2018).

<https://doi.org/10.1007/s11663-018-1172-4>

RE-fluoride [Okabe_2018_b]

Okabe, T. H., Taninouchi, Y. & Zheng, C. Thermodynamic Analysis of Deoxidation of Titanium Through the Formation of Rare-Earth Oxyfluorides. *Metall. Mater. Trans. B* **49**, 3107–3117 (2018).

<https://doi.org/10.1007/s11663-018-1386-5>

Experimental work:

Y-chloride [Iizuka_2020]

Iizuka, A., Ouchi, T. & Okabe, T. H. Ultimate Deoxidation Method of Titanium Utilizing Y/YOCl/YCl₃ Equilibrium. *Metall. Mater. Trans. B* **51**, 433–442 (2020).

<https://doi.org/10.1007/s11663-019-01742-6>

La-chloride [Tanaka_2020]

Tanaka, T., Ouchi, T. & Okabe, T. H. Lanthanothermal Reduction of TiO₂. *Metall. Mater. Trans. B* **51**, 1485–1494 (2020).

<https://doi.org/10.1007/s11663-020-01860-6>

Ce-chloride [Kamimura_2022]

Kamimura, G., Ouchi, T. & Okabe, T. H. Deoxidation of Titanium Using Cerium–Chloride Flux for Upgrade Recycling of Titanium Scraps. *Mater. Trans.* **63**, 893–902 (2022).

<https://doi.org/10.2320/matertrans.M-M2022805>

Ho-chloride [Kong_2021]

Kong, L., Ouchi, T. & Okabe, T. H. Deoxidation of Ti using Ho in HoCl₃ flux and determination of thermodynamic data of HoOCl. *J. Alloy. Compd.* **863**, 156047 (2021).

<https://doi.org/10.1016/j.jallcom.2020.156047>

The activities of the deoxidant (Y), flux (YF₃), and reaction product (YOF) were set to unity in principle. For the decrease in Y activity due to its dissolution in the Ti melt, the calculated results with varying Y activity are presented in Figs. 2a and 2c. Given the substantial addition of YF₃ flux in the experiments and the limited solubility of yttrium oxyfluoride in the liquid YF₃ phase near the melting point of Ti [Baek_2022], it was considered reasonable to set the activities of YF₃ and YOF to unity.

The vapor pressures of metallic Y and YF₃ at 2000 K (1727 °C) are relatively low, measuring 3.97×10^{-5} atm and 7.77×10^{-3} atm, respectively [1991_Knacke]. This suggests that Y and YOF likely remained in their condensed phase without evaporating during the melting process with Ti, which lasts about 15 minutes.

The current analysis concerns the thermodynamic relationship between Y and oxygen concentrations in the Ti melt during the deoxidation equilibrium forming YOF, as illustrated in Fig. R1. Despite the significant margin of error, the deoxidation limit of the Ti melt in the Y/YOF/ YF₃ equilibrium at 2000 K (1727 °C) is predicted to be between 100 and 300 mass ppm O under the conditions of Experiment #1. Notably, the experimental results in the current work, as plotted in Figure R1, align well with this prediction. However, due to the limited mutual solubility between solid Ti and Y, Y segregation which occurred during the cooling in these experiments was observed in the solidified Ti samples. This segregation complicates the accurate determination of the true

concentration of dissolved Y in the molten Ti. As a result, the experimental results shown in Fig. R1 are provisional, and discussions based on these findings are not included in this paper or the Supplementary Information.

REDACTED

Fig. R1. Thermodynamic relationships between Y and oxygen concentrations in Ti–Y–O melt coexisting with solid Y_2O_3 phase or two phases of liquid YF_3 and solid $Y_4O_3F_6$ at 2000 K (1727 °C). (Unpublished data: Since it relates to intellectual property, such as know-how and experimental conditions, this figure should be confidential and undisclosed information at this time.)

The following explanation was added in the manuscript after L. 138 and after L. 152. The thermodynamic data utilized are summarized in Supplementary Table 1 in the Supplementary Information.

After L. 138

Thermodynamic data for compounds in the Y–O–F system at 2000 K (1727 °C) are compiled in Supplementary Table 1. Thermodynamic data for Y_2O_3 and YF_3 were based on the values tabulated in Barin's compilation of thermodynamic data²⁴. Baek and Jung's work shows that the liquid YF_3 phase can coexist with the yttrium oxyfluoride phases at temperatures as high as 1973 K (1700 °C), surpassing the melting point of Ti. Thermodynamic data for oxygen in liquid Ti at 2000 K (1727 °C) are also included in Supplementary Table 1. These data are sourced from various sources scattered throughout the literature and are characterized by significant uncertainty.

After L. 152

In the current work, the activities of the deoxidant (Y), the flux (YF_3), and the reaction product (YOF)—with reference states corresponding to pure solid Y, pure liquid YF_3 , and pure solid YOF, respectively—were set to unity in principle. Calculations accounting for the decrease in Y activity due to its dissolution into the Ti melt are detailed in Figures 2a and 2c. Given the substantial amount of YF_3 flux used in the experiments and the relatively low solubility of yttrium oxyfluoride in the liquid YF_3 phase near Ti's melting point²⁵, it was deemed reasonable to assume unity for the activities of YF_3 and YOF.

2. For the thermodynamic calculations, what data sources were used for properties of the Y-O-F system at elevated temperatures? Were these extrapolated? This affects the uncertainty.

As mentioned above in response to Comment 1, thermodynamic data for YOF at around 1273 K (1000°C) were sourced from the studies by Levitskii and Balak (Ref. 26) [Levitskii_1982] and Baek and Jung (Ref. 25) [Baek_2022]. Lower temperature data was extrapolated to estimate the high-temperature data at approximately 2000 K (1727°C), which is above the melting point of Ti. Other data were derived from Barin's compilation of thermochemical data (Ref. 24) [Barin_1995].

Despite the significant margin of error, the calculations predicted a deoxidation limit for Ti in the Y/YOF/ YF_3 equilibrium at 2000 K (1727°C) to be between 100 and 300 mass ppm O under the conditions of Experiment #1. These predictions closely align with the experimental results obtained in the current work.

3. The experimental results are promising, but measurements of additional samples would make the conclusions more robust. With only 1-2 samples for most conditions, variability between samples is unclear.

For Experiment #1, 10 specimens were obtained from different areas within the Ti sample to measure the oxygen concentration. As illustrated in Figure R1, the oxygen concentration uniformly decreased from the interior to the exterior and from the top to the bottom of the sample, reaching approximately 200 mass ppm. This uniform reduction indicates that deoxidation occurred throughout the entire sample. Additionally, in 10 out of 13 trials conducted under various conditions, initial samples with oxygen concentrations of 1000 mass ppm and 10000 mass ppm were deoxidized to below

1000 mass ppm O, demonstrating the reproducibility of the deoxidation process. Typically, simply melting Ti does not reduce the oxygen concentration; it invariably increases due to oxygen contamination from the gas atmosphere or the crucible material.

Fig. R2. Distribution of oxygen and nitrogen concentration in the Ti sample of Exp. #1.

Oxygen and nitrogen concentrations were analyzed at 10 points across 6 locations, indicated by black dots. The results are summarized in the table. (The initial oxygen concentration in the Ti sample was approximately 1000 mass ppm.) Across all analysis points, including the center and near the bottom of the sample, the oxygen concentration consistently decreased to around the 200 mass ppm level. This uniform reduction throughout suggests that the deoxidation reaction occurred evenly across the entire sample.

*4. Agitation of the melt is mentioned but not characterised. Mixing likely plays a role in deoxidation rate and uniformity.
Quantifying mixing time/velocity would aid understanding.*

The induction skull melting (ISM) furnace utilizes high frequencies not only for heating, but also for stirring and levitating the melt. In these experiments, it is likely that the titanium melt was vigorously stirred due to the flow of induction current through it. Indeed, upon visual observation during the experiment, the surface of the melt was seen to be extremely fluid. For a Ti sample of approximately 300 grams, depending on the raw materials and the output and frequency of the power source used for melting, complete melting and stirring could be achieved in as little as 10 minutes. In the experiments in the current work, the melting process lasted between 10 and 20 minutes, indicating the likelihood that the molten Ti was uniformly mixed. This uniformity in mixing is corroborated by the oxygen analysis results shown in Fig. R2, which indicate that the Ti melt was thoroughly stirred and uniformly dissolved. Moreover, with intense stirring under the high-temperature conditions at 2000 K (1727 °C), it is believed that the reaction rate of deoxidation and the mass transfer of reactive species were extremely rapid, bringing the deoxidation reaction close to equilibrium.

To inform readers that the experiment was conducted under strong agitation conditions, we added the following statement in Line 338:

→ “Visual observation confirmed that the molten Ti was strongly agitated by electromagnetic forces.”

5. In Table 1, please add units for the materials charged (e.g., g).

→ Materials charged into induction skull melting, w_i /g

The table currently indicates the quantities of Ti, rare-earth metals, and fluorides (w_i) in grams (g), as above. If this is unclear, would it be preferable to display the units within parentheses like this?

→ Materials charged into induction skull melting, w_i (g)

6. How was mixing/agitation of the molten Ti characterised? Is it well mixed to enable uniform oxygen removal?

As noted in the response to comment 4, the ISM furnace is used not only for heating through high frequency but also for stirring and levitating the melt. The flow of induction current through the Ti melt suggests that it was vigorously agitated during the melting experiment. Indeed, visual observations during the experiment indicated that the melt's surface was highly fluid. For a Ti sample of around 300 g, complete melting and agitation are typically achieved within 10 minutes, depending on the raw material used and the settings of the power supply used for melting. In the experiments in the current work, where the melting time was 10 to 20 minutes, the deoxidation reaction is believed to have approached equilibrium.

To convey to the readers that the experiments were conducted under conditions of strong agitation, a description has been added to Line 338.

→ “Visual observation confirmed that the molten Ti was strongly agitated by electromagnetic forces.”

Additionally, Supplemental Information has been included to demonstrate that oxygen concentrations were uniformly distributed across the entire sample.

7. The integrated refining process in Fig. 6 is interesting but remains conceptual. Significant engineering development is needed to implement this commercially. The challenges should be discussed.

Your observation is entirely correct. In the current work, an ISM furnace was employed to facilitate rapid deoxidation equilibrium through the uniform agitation of molten Ti. However, for commercial applications, a larger melting furnace, like a plasma arc furnace, might be more appropriate. To ensure clarity for readers on this point, we have included the following explanation from Line 288:

- “The deoxidation method for molten Ti demonstrated in the current work is applicable in various types of melting furnaces. For melting large quantities of Ti on an industrial scale, a larger-scale furnace, such as a plasma arc furnace, would be better suited than the induction skull furnace used in the current work.”

8. The recycling of rare earth metals is mentioned but not elaborated on. Details would strengthen the economic/sustainability case.

As described below, Y, La, and Ce are produced in large quantities as by-products of Nd, Dy, and Tb, which are used in magnets. Therefore, when using this method for small-scale titanium production or recycling, it is not necessarily required to recycle rare-earth metals. However, as you note, if this method were to be applied in the future for large-scale titanium (Ti) production, it would indeed be important to recycle reaction products like oxyfluorides (YOF) to produce Y and YF₃ for reuse. It is technically feasible to recycle these elements with existing technologies capable of converting YOF into YF₃ and extracting metallic Y from YOF by molten salt electrolysis.

Line 243

- “In recent years, the increased production of magnet materials such as neodymium (Nd), dysprosium (Dy), and terbium (Tb) has resulted in an oversupply of Y, La, and Ce, which are simultaneously produced as by-products^{28–33}. As a result, processes that utilize these abundant and surplus rare earth metals are emerging as economically rational and resource-efficient options.”

Furthermore, in Line 278, the following clarification has been added for readers to show that Y metal and YF₃ flux can be recycled from YOF with existing technology.

- “Furthermore, it is also technically possible to regenerate Y metal and YF₃ from YOF using methods such as molten salt electrolysis.”

9. The Conclusions section focuses narrowly on the novel deoxidation approach but could be broadened to the larger implications for Ti production.

As you point out, the technique developed in the current work has the potential to offer a future alternative to the Kroll process, the current method used for the mass production of Ti.

In Line 307, we state:

➔ This technique holds the potential to establish an innovative refining process for manufacturing high-purity Ti directly from TiO_2 , eliminating the need for chloride refining. Furthermore, this technique also enables the upgrade recycling of high-oxygen-concentration Ti scrap.

To expand on this, we have added the following statement in Line 310:

Line 316

➔ In future, this method has the potential to become a new technique for producing Ti, thereby offering an alternative to the Kroll process, the current standard for Ti mass production.

10. How was YOF stability characterised at high temperature? What indications are there that it persists to 2000K?

In the current work, assuming YOF's stability at high temperatures, the deoxidation limit of Ti by Y was predicted and then validated experimentally (see Fig. 2 in the article). The study by Baek and Jung on the experimental measurement of the Y_2O_3 – YF_3 pseudobinary phase diagram reveals that the YF_3 liquid phase can be in equilibrium with yttrium oxyfluoride phases at temperatures as high as 1973 K (1700 °C), which is above the melting point of Ti (see Fig. R3 below). The experimental results, showing Ti with oxygen concentrations consistent with predictions, confirm YOF's high-temperature stability or a significant reduction in Y_2O_3 's apparent activity in the reaction system due to its dissolution into the flux.

Fig. R3. Phase diagram for the Y_2O_3 – YF_3 pseudobinary system.
(S. Baek and I.-H. Jung, J. Eur. Ceram. Soc. 42, 5079 (2022).
<https://doi.org/10.1016/j.jeurceramsoc.2022.05.005>)

11. Were the samples crafted into test specimens to measure mechanical properties and validate usability?

At this time, we have not measured the mechanical properties of the deoxidized Ti. In fact, Ti deoxidized to a level of 200 mass ppm O is very soft, indicating a low concentration of interstitial elements.

12. How rapidly does deoxidation occur? Is there an optimum mixing speed?

In the current work, a melting/deoxidation experiment was conducted under conditions where, following visual confirmation that the Ti sample had completely melted, the Ti melt was subjected to intense stirring by electromagnetic induction for over 10 minutes. The results indicated that the oxygen concentrations in Ti samples from different locations were almost uniformly consistent and aligned closely with thermodynamic predictions. This suggests that the deoxidation reaction reached equilibrium within 15 minutes under these experimental conditions. Although deoxidation might occur in even shorter periods of less than 15 minutes, the minimum time needed to achieve equilibrium was not measured in the current work.

13. What level of rare earth contamination in the Ti is acceptable for various applications?

We cannot provide an exact answer to your question, as the acceptable concentrations of rare-earth metals in standard Ti materials and alloys are not strictly defined. Ti produced using the current method may not be suitable for high-reliability Ti alloys such as those used in aircraft. However, we believe that Ti containing a certain amount of rare-earth elements could be viable for use in everyday consumer products. The results of Experiment #2 showed that the concentration of rare-earth elements (Y) in the produced Ti was below 0.1 mass% (1000 ppm). At the same time, it was confirmed that the oxygen level in Ti could be successfully reduced to less than 600 mass ppm under these conditions.

14. What prior attempts have been made at direct oxide-to-metal smelting processes? Why did they fail or not advance?

Past attempts have been made to produce metallic Ti directly from TiO₂ using Ca and rare-earth metals as reducing and deoxidizing agents, including in our own work. Some of these efforts have successfully demonstrated the production of solid Ti with oxygen concentrations below 500 mass ppm at around 1300 K (1027 °C). However, at this temperature, the diffusion of oxygen in solid Ti is extremely slow, hindering the possibility of mass-producing Ti metal quickly. Moreover, as indicated in Ref. 9 [Kobayashi_1998] and Ref. 10 [Fukuzawa_1999], it was believed that direct deoxidation of the Ti melt to levels below 1000 mass ppm was unfeasible due to the reduced deoxidation capabilities of Ca and rare-earth metals at higher temperatures. More recent research, including our own, has found that using the formation reaction of oxyfluorides such as YOF significantly lowers the deoxidation limit in solid Ti, potentially to below 100 mass ppm. Furthermore, given recent work by other

researchers demonstrating that oxyfluorides such as YOF are stable even above the melting point of Ti, the current work has demonstrated for the first time that oxygen can be directly removed from Ti melt to the level of 200 mass ppm O.

15. Define acronyms first mention (e.g. ATR)

ATR and MTR are described on page 3, line 51, as follows.

Line 51

➔ *These processes are known as the aluminothermic reduction (ATR) and the magnesiothermic reduction (MTR)²⁻⁶,*

16. Avoid directional language like "this study" - replace it with "the current work"

We have changed "this study" to "the current work" in the paper based on your suggestion. The correction is made on lines 15, 96, 98, 116, 120, 128, 152, 185, 188, 202, 279, 285, 287, 290, 301, etc.

17. Reduce the use of "we"

We have reduced the use of "we" in the paper based on your suggestion. The correction is made on lines 15, 19, 66, 99, etc.

18. Tighten wording in places for brevity and clarity

We have tightened wording in places for brevity and clarity based on your suggestion. The correction is made on line 267, 270, 276, 301, etc.

19. Check references to match journal style

We have checked references to match the journal style. The correction is made on Refs. 11, 28, 29, 30, and 33.

20. Abstract: Change "Endowed with superior mechanical and chemical properties" to "Possessing superior mechanical and chemical properties."

The correction has been made on line 12 based on your suggestion.

21. Line 53: Change "invariable retains" to "invariably retains"

The correction has been made on line 59 based on your suggestion.

22. Line 81: Change "imbuing" to "introducing"

The correction is made on line 170 based on your suggestion.

23. Line 347: Add a space between "J58719A1" and the comma

24. Add Oxford comma in a few places with lists of 3 items

The correction is made on line 385 based on your suggestion. All lists of three items have been checked to ensure consistent use of the Oxford comma.

In summary, while the deoxidation approach shows promise, the paper would benefit from strengthening the thermodynamic and experimental sections, better contextualising the proposed process, and expanding the conclusions.

We acknowledge the importance of precise thermodynamic predictions and detailed sample evaluation, especially given the challenges in conducting experiments at temperatures around the melting point of Ti. As you have pointed out, the thermodynamic data on yttrium oxyfluoride (YOF) at high temperatures are not very reliable. However, the good alignment of our theoretical calculations with the deoxidation experimental results supports the validity of our approach. We have made additions and corrections to both the Thermodynamics and Experimental sections. Furthermore, to enhance the reader's understanding of the validity of our results, the Supplementary Information now includes experimental data on the uniformity of oxygen concentration in the obtained Ti samples and on the presence and stability of YOF.

Thank you very much for your highly constructive and appropriate guidance and suggestions. Your comments will be invaluable not only for the current work but also for our future research endeavors. We are deeply grateful.

Reviewer #2 (Remarks to the Author):

Review of T. H. Okabe et al.: „Direct Production of Low-Oxygen-Concentration Titanium from Molten Titanium”

General comment on authors:

T. H. Okabe has published many important articles on alternative methods for titanium metal extraction during the last almost twenty years, e.g. on calciothermic and magnesiothermic reduction, usage of halides and electrochemical methods. In 2019 and 2020, respectively, he authored two review papers in this field and a book chapter (reference 1 in current work). Recently, he published several articles on the usage of rare earths in combination with halide fluxes, as kind of predecessors to the current work.

The co-authors have published several articles on extractive metallurgy of titanium as well. Therefore, it is safe to say that the authors of the current work have got a high expertise in this field.

General comments on the approach described in current work:

The key aspect of the approach is the deoxidation of titanium metal by rare earth metals (e.g. yttrium Y) and a halide flux (e.g. yttrium fluoride YF_3) with the formation of oxyhalides (e. g. YOF) instead of corresponding oxides, thus achieving significantly lower oxygen-contents.

The novelty of this approach is unquestionable.

The soundness of this approach is made plausible by suitable thermodynamic calculations and lab-scale experiments.

The paper is in principle suitable for publication in Nature Communications.

General critics/suggestions:

• **Contamination of the deoxidized titanium by yttrium** is significant, as evident from the current work and several preceding publications. This issue is not sufficiently addressed in the current work.

Experimentally obtained Y-contents are only roughly stated in the text, but instead they should be precisely given in Table 1 together with the contents of oxygen and nitrogen. In this way, the varying oxygen-contents become also more understandable (solubility product of Y and O, according to equation 3 and 4). According to section “post-experiment analysis”, Y and the other rare earth elements have been analyzed by ICP-AES and XRF, thus their exact contents should be given in Table 1. In addition, a general remark could be appropriate that the consequences of significant contents of the rare earths in titanium on its properties in the final application are not yet sufficiently investigated to date.

You have correctly pointed out that Table 1 should include the concentrations of Y, La, and Ce, alongside the amounts of oxygen and nitrogen. However, Y, La, and Ce have limited solubility in solid Ti, leading to significant segregation during cooling in the experiments. Determining the exact concentration of these rare-earth elements in molten Ti remains a challenge. We are working diligently to quantify the dissolved Y concentration accurately and are in the process of validating these findings, as illustrated in Figure R1. Some experiments have yielded Ti with Y concentrations under 1000 mass ppm and oxygen levels also below 1000 mass ppm. This demonstrates that Ti can be effectively deoxidized in conditions that minimize Y contamination, a critical consideration for its industrial use.

Moreover, as you have also highlighted, the influence of rare-earth elements on the properties of Ti for end-use applications has not yet been fully investigated. The low-oxygen Ti produced by the current work may not be suitable for high-reliability applications such as aerospace-grade Ti alloys. Nevertheless, we are optimistic about the potential application of this Ti in consumer products, such as a material in devices like iPhones.

REDACTED

Fig. R1. Thermodynamic relationships between Y and oxygen concentrations in Ti–Y–O melt coexisting with solid Y_2O_3 phase or two phases of liquid YF_3 and solid $Y_4O_3F_6$ at 2000 K (1727 °C). (Unpublished data: Since it relates to intellectual property, such as know-how and experimental conditions, this figure should be confidential and undisclosed information at this time.)

• Temperature control in the experiments is not clear. Induction melting, in particular with a cold crucible, generally is challenging regarding temperature control of the melt. The temperature in turn is crucial to the thermodynamic equilibrium of the deoxidation (e. g. equation 4). It should be stated how the temperature was measured (radiation pyrometer, thermocouple etc.) and which values or range were obtained.

We did not use a radiation pyrometer or thermocouple for direct temperature measurement. However, the cold crucible induction melting furnace (induction skull melting furnace) used in this experiment is thought to have heated only to about 50°C above the melting point of the material. This is due to the large contact area between the water-cooled copper crucible and the molten Ti (350 g) and the presence of solid Ti at the interface. Previous measurements indicate that the furnace does not

heat the metal to more than 100°C above its melting point. This is because the stirring of the melt and the resulting radiative heat loss are substantial relative to the power input.

*• **Application of cold crucible (“skull”) induction melting** possibly limits the potential for industrial scale-up and is very energy intensive. However, it is clear from the current state of the art and science that suitable refractory crucibles have not been found yet for high-Ti melts despite decades of intensive research and development. Y₂O₃ has been applied for Ti-melts in prior publications but would not be suitable in the current approach due to the need for saturation of the flux with Y₂O₃ in that case. So, it remains open what refractories could be used in the process sketched in Figure 6. Nevertheless, compared to conventional melting processes for titanium (vacuum arc remelting, electron melting and plasma arc melting), cold crucible induction melting could still be advantageous.*

In the current work, a cold crucible (induction skull melting) furnace was used, primarily to facilitate the rapid melting of the experimental sample, expedite the deoxidation reaction through the magnetic levitation and vigorous agitation of the Ti melt, and ensure the uniformity of the sample.

You are correct in pointing out that the application of cold crucible (“skull”) induction melting does have its limitations for industrial scale-up. Currently, ISM furnaces capable of melting approximately 100 kg per batch are in practical use,

However, as you note, for melting on the order of tons, plasma arc melting furnaces and electron beam melting furnaces would be more appropriate.

In scaling up and commercializing the process depicted in Figure 6, one viable approach might be to employ a plasma torch or a similar heating method that heats the Ti melt from above, especially when using a water-cooled copper crucible. Alternatively, using high-melting-point metals like Mo and W, which require external cooling, could also be considered. We are currently investigating the feasibility of using solid rare-earth fluorides or oxyfluorides as refractory linings in reaction vessels.

There are the following comments and suggestions:

- **Equations (3):** One should assume that the RE is dissolved in the Ti and that RE_X and REOX form a homogeneous mixture phase. At 1700°C Ti and Y are completely miscible. Then, the equation reads:

As a consequence, the reduced activities of the corresponding component have to be considered in the thermodynamic calculations.

- **Figure 2b:** E.g. according to the reaction

the mole fraction of x_O in Ti is given by

$$x_O = \frac{1}{\gamma_O} \left(\frac{K(T) a_{Y_2O_3}}{a_Y^2} \right)^{1/3}$$

with K(T) equilibrium constant and γ_O the activity coefficient of O in Ti (which in principle also depends on x_O). The equation can also be written as

$$P_{O_2} = a_O = \left(\frac{K(T) a_{Y_2O_3}}{a_Y^2} \right)^{1/3}$$

(O₂ reference state for oxygen). So, the position of the hollow circles in Figure 2b depend on a_Y (and thereby on x_Y) and on a_{Y₂O₃}. Of course, the same is also the case for the other symbols. It is o.k. to make a reasonable choice for the activities (e.g. a=1) but it should be explained more clearly which one and why.

It is o.k. to make a reasonable choice for the activities (e.g. $a=1$) but it should be explained more clearly which one and why.

To assess the decrease in Y activity due to its dissolution in the Ti melt, calculation results with varying Y activities are shown alongside each other in Figs. 2a and 2c. To our knowledge, there are no reported measurements of the activities of Ti and Y in the Ti-Y binary system's liquid phase. Given that a substantial amount of YF₃ flux was added in the experiments, and the solubility of yttrium oxyfluoride in the liquid YF₃ phase is relatively low near the melting point of Ti, as cited in (Ref. 25) [Baek_2022], setting the activities of YF₃ and YOF to unity is considered reasonable. The analysis of Figure R1 suggests the absence of any significant discrepancy between our thermodynamic analysis and the experimental results.

We have added the following explanation after Line 152 in the thermodynamic calculation section of the manuscript:

From Line 152

- ➔ "...the current work. The activities of the deoxidant (Y), the flux (YF₃), and the reaction product (YOF)—with reference states corresponding to pure solid Y, pure liquid YF₃, and pure solid YOF, respectively—were set to unity in principle. Calculations accounting for the decrease in Y activity due to its dissolution into the Ti melt are detailed in Figures 2a and 2c. Given the substantial amount of YF₃ flux used in the experiments and the relatively low solubility of yttrium oxyfluoride in the liquid YF₃ phase near Ti's melting point²⁵, it was deemed reasonable to assume unity for the activities of YF₃ and YOF."

Minor comments and suggestions:

- *It's good practice to mention the reference state when using activities like in Figure 2a.*

The following explanation was added from L. 152 in the thermodynamic calculation section of the manuscript.

- ➔ "In the current work, the activities of the deoxidant (Y), the flux (YF₃), and the reaction product (YOF)—with reference states corresponding to pure solid Y, pure liquid YF₃, and pure solid YOF, respectively—were set to unity in principle."

- *Line 45 etc.: A further significant disadvantage of the conventional extraction route is the need for remelting the titanium sponge from Kroll process, thus increasing costs and energy demand significantly.*

Thank you for your comments on the revision and improvement of the text. We have added the above statement to the text at Line 43–45 as follows:

➔ A further significant disadvantage of the conventional extraction route is the need to melt the Ti sponge produced from the Kroll process, which significantly increases costs and energy consumption.

• *Line 48: Expression “...both of which exhibit higher chemical affinities with oxygen than Ti” is imprecise and not quite right. As also evident from Figs. 2b and 2c, due to the significant oxygen solubility of titanium, the oxygen potential pO_2 required for reduction/deoxidation is delicately determined by the pursued oxygen content.*

You are correct. The text has been revised as follows. “...both of which exhibit higher chemical affinities with oxygen than Ti” on Lines 50–51 has been replaced with “...both of which exhibit strong chemical affinities with oxygen.”

• *Line 60 etc.: Regarding aluminothermy, the significant contamination of titanium with aluminium could be mentioned, according to the solubility product in the system Ti-Al-O. Furthermore, the seminal work of Maeda et al. (reference 22) could also be cited here.*

As you suggested, we have included a reference to Prof. Maeda's work in our discussion of ATR-Ti production (Line 52) and have added the following text after Line 62:

➔ “Furthermore, ATR-Ti is heavily contaminated with Al, rendering it unsuitable for commercial applications in its current state⁷.”

• *Line 60 etc.: Why is calciothermy not mentioned beside aluminothermy and magnesiothermy? Due to costs for calcium?*

Yes, we did not include calcium (Ca) in our discussion due to its prohibitively high production costs. In the past, we have conducted research on the reduction of TiO_2 using metallic Ca and developed a process that could produce metallic Ti with a low oxygen concentration of around 500 mass ppm, as well as minimal Ca contamination. However, we found that using Ca as a reductant is not economically feasible since a technology for achieving the low-cost production of metallic Ca has yet to be established.

• *Line 183: “... while suppressing the incorporation of Y into the deoxidized Ti melt”. It would be better to write “... while minimising the incorporation of Y into the deoxidized Ti melt”. A “suppressing” of Y-pick up is thermodynamically impossible.*

Thank you for pointing this out. The manuscript has been updated on Line 208 make the suggested revision.

• *Line 200 (Fig. 3): Although the apparatus dimensions are later given in the section “experimental setup”, why not draw the internal diameter of the crucible or a scale right in the sketch?*

Thank you for pointing this out. We will update the manuscript to make the suggested revision.

• Line 210 (Fig. 4c): Is the outer “ring” of the sample the solidified flux? (could be mentioned in the corresponding caption)

As you have intuited, the outer “ring” in the image is indeed solidified flux. Acknowledging your observation that Fig. 4c is not very clear, we have replaced it with a new photo that clearly marks the metallic Ti and the flux.

Fig. 4c.

• *Figure 4d: The Figure is difficult to understand.*

As you have pointed out, Fig. 4d is indeed difficult to understand, so we have replaced it with the following photo. The oxygen analysis results shown on the left in the photo (Fig. R2) are provided for reference only.

Fig. 4d.

Fig. R2. Distribution of oxygen and nitrogen concentration in the Ti sample of Exp. #1.

Oxygen and nitrogen concentrations were analyzed at 10 points across 6 locations, indicated by black dots. The results are summarized in the table. (The initial oxygen concentration in the Ti sample was approximately 1000 mass ppm.) Across all analysis points, including the center and near the bottom of the sample, the oxygen concentration consistently decreased to around the 200 mass ppm level. This uniform reduction throughout suggests that the deoxidation reaction occurred evenly across the entire sample.

• *Line 218 (Table 1): The last column (“Experimental code”) contains no information relevant to the public, unless it refers to supplementary data additionally provided. It would be very useful to give also the Y-, La-, Ce-content in the final material.*

You are correct in noting that the “Experimental code” column, while important for our experimental management, holds no significance for the readers. Accordingly, it has been removed.

Regarding your suggestion to include concentrations of Y, La, and Ce in Table 1 along with the oxygen and nitrogen contents, we agree with you in principle. However, unlike oxygen, Y, La, and Ce are only minimally soluble in solid Ti, leading to significant segregation during the cooling phase of the experiments. Currently, we are working to quantify the concentrations of these rare-earth elements dissolved in the Ti melt in more precise terms and to assess their reliability, as illustrated in Figure R1. In some experiments, we obtained Ti samples with Y concentrations of less than 1000 mass ppm and oxygen concentrations also below 1000 mass ppm. Demonstrating that Ti can be effectively deoxidized under conditions that minimize Y contamination is a key insight for the industrial application of deoxidized Ti.

• Line 260 (Fig. 6): Designation “Flux (CaF₂, etc.)” could be extended by “YF₃”, according to the approach and findings of the current work. What are the designated refractory materials?

Thank you for your suggestion. We have accordingly made the following revision in Fig. 6:

➔ Flux (CaF₂, etc.) ➔ Flux (YF₃, CaF₂, etc.)

In scaling up and commercializing the process depicted in Figure 6, one viable approach might be to employ a plasma torch or a similar heating method that heats the Ti melt from above, especially when using a water-cooled copper crucible. Alternatively, using high-melting-point metals like Mo and W, which require external cooling, could also be considered. We are currently investigating the feasibility of using solid rare-earth fluorides or oxyfluorides as refractory linings in reaction vessels.

• Line 296: Pressure of argon atmosphere (e.g., “ambient”)?

Thank you for your observation. The experiments were conducted in an Ar atmosphere at a pressure of 500 Torr. We will update the manuscript to make the suggested revision. See Line 332.

• Line 356: “...seoxidation” (typo).

Thank you for pointing this out. We will update the manuscript to make the suggested revision. See Line 394.

Thank you very much for your highly constructive and appropriate guidance and suggestions. Your comments will be invaluable not only for the current work but also for our future research endeavors. We are deeply grateful.

Reviewer #3 (Remarks to the Author):

This paper presents a method and experimental examples of the deoxidation of molten titanium.

Merits:

- 1. Deoxidation is a very challenging problem for Ti. The low oxygen content, <200 ppm, presented in the article is remarkable.*
- 2. The process presented could be used for the production of primary Ti as well as the deoxidation of scrap Ti.*

Detractors:

- 1. The novelty of the presented work is questionable. The authors' group has been very active in this area. I easily found, using Google search, over ten publications by the group on the general subject of using rare earth elements, specifically Y, to deoxidate Ti. Some are for deoxidate solid Ti, some molten Ti. I tried to see if the use of flux to form Y oxychloride is novel, but, the following publication seems to have significant similarity.*

TY - JOUR

AU - Okabe, Toru

AU - Taninouchi, Yu-ki

AU - Zheng, Chenyi

PY - 2018/08/31

SP -

T1 - Thermodynamic Analysis of Deoxidation of Titanium Through the Formation of Rare-Earth Oxyfluorides VL - 49 DO - 10.1007/s11663-018-1386-5 JO - Metallurgical and Materials Transactions B

As you have pointed out, we have conducted a series of studies over the years focused on removing oxygen from *solid* Ti. These studies primarily involved extracting oxygen from the surface of solid Ti at temperatures around 1300 K (1027 °C) by utilizing diffusion within the solid Ti. It has been demonstrated that Ti with oxygen concentrations below 100 mass ppm can be produced at around 1300 K (1027 °C).

In contrast, the current work introduces a novel method for directly removing oxygen from *liquid* Ti at high temperatures above its melting point. The deoxidation of liquid Ti has proven both thermodynamically and technically challenging, and the general understanding until now has been that producing Ti with oxygen levels below 1000 mass ppm by deoxidating Ti in a molten state was not feasible, as indicated in Ref. 9 [Kobayashi_1998] and Ref. 10 [Fukuzawa_1999], among others.

We believe the new technique developed in the current work, which is capable of directly producing Ti with low oxygen concentrations of around 200 mass ppm from the melt, represents a significant breakthrough.

- 2. The presentation of the article is a little unconventional. The authors presented examples number 1, 2, 3, etc. It reads like a patent application. The style makes it difficult to grasp the method's principles beyond the data report.*

In response to your comment, we have revised the table, changing “No.” to “Exp. #” to ensure consistency with the experimental descriptions elsewhere in the text (e.g., Exp. #1, Exp. #2). Additionally, based on feedback from another reviewer, we have removed the “Experimental code” from the table.

3. Another very minor point. XRD cannot be used to determine if residual oxides of other elements are present. It is not sensitive enough.

In response to your point about the limitations of XRD, we noted that the XRD measurement of the flux displayed distinct diffraction peaks for yttrium oxyfluoride that differed from those of Y and YF_3 . This led us to conclude that YOF and other oxyfluorides formed after the deoxidation experiments. While it is true, as you mentioned, that XRD cannot provide a quantitative assessment of Y_2O_3 , we observed no diffraction peaks of Y_2O_3 in the flux. Additionally, SEM analysis of the fluxes confirmed the presence of oxyfluorides but did not detect Y_2O_3 .

To clarify this point further, we have added enlarged views of the XRD patterns in Fig. 5b to show the absence of Y_2O_3 in the flux. We have also included the SEM-EDS results of the fluxes in Supplementary Figure 6 and Supplementary Table 5.

Fig. 5
 XRD patterns of YF_3 fluxes before and after deoxidation. The intensities are shown in (a) wide and (b) narrow diffraction angle scales. Various yttrium oxyfluoride phases were detected in the fluxes after the deoxidation experiments, whereas no yttrium and titanium oxide phases were observed, which shows that the $\text{Y}/\text{YOF}/\text{YF}_3$ equilibrium was established in the current systems.

Thank you very much for your highly constructive and appropriate guidance and suggestions. Your comments will be invaluable not only for the current work but also for our future research endeavors. We are deeply grateful.

The following are items that were not directed by the reviewers, but the items that the authors felt the need to revise upon reviewing.

Line 232

agitated well by pinch force with induction heating,

=>

agitated well by Lorenz force (pinch force) with induction heating,

Line 234

subjected to the pinch force. => subjected to the Lorenz force.

Line 245

due to induction melting. => after induction melting.

Line 426

Mr. Yoichi Takashina of Toho Technical => Mr. Yoichi Warashina of Toho Technical

The method used for the levitation and agitation of the Ti melt during the melting of Ti samples in the current work is an important technique. The slit in the water-cooled copper crucible, which facilitates the introduction of high-frequency waves as depicted in the original Fig. 4a of the apparatus, is currently considered confidential and undisclosed information due to its association with intellectual property, such as experimental know-how and conditions. Consequently, in the revised Fig. 4a, the photograph of the slit in the water-cooled copper crucible has been modified with blurring for image confidentiality.

REDACTED

Photographs of a Ti sample after melting and deoxidation using a cold crucible induction melting furnace (Exp. #1 in Table 1). a, The sample in the cold crucible after melting; b, the side and c, top views; and d, Sectioned Ti sample pieces for analysis. The metal sample was separated from the flux phase after induction melting.

*** See Nature Portfolio's author and referees' website at www.nature.com/authors <<http://www.nature.com/authors>> for information about policies, services and author benefits.*

This email has been sent through the Springer Nature Tracking System NY-610A-NPG&MTS.

Confidentiality Statement:

This e-mail is confidential and subject to copyright. Any unauthorized use or disclosure of its contents is prohibited. If you have received this email in error please notify our Manuscript Tracking System Helpdesk team at <http://platformsupport.nature.com>

Details of the confidentiality and pre-publicity policy may be found here <http://www.nature.com/authors/policies/confidentiality.html>

Privacy Policy <<http://www.nature.com/info/privacy.html>> | Update Profile <<https://mts-ncomms.nature.com>>

DISCLAIMER: This e-mail is confidential and should not be used by anyone who is not the original intended recipient. If you have received this e-mail in error please inform the sender and delete it from your mailbox or any other storage mechanism. Springer Nature America, Inc. does not accept liability for any statements made which are clearly the sender's own and not expressly made on behalf of Springer Nature America, Inc. or one of their agents.

Please note that neither Springer Nature America, Inc. or any of its agents accept any responsibility for viruses that may be contained in this e-mail or its attachments and it is your responsibility to scan the e-mail and attachments (if any).

REVIEWERS' COMMENTS

Reviewer #1 (Remarks to the Author):

Based on my review, the authors have thoroughly addressed the reviewers' comments and suggestions to improve the manuscript. Here are some of the key changes made:

1. More details on the thermodynamic calculations, data sources for Y-O-F system properties at high temperatures, and assumptions on activity coefficients were added. This strengthens the analysis.
2. Clarified that the titanium melt was strongly agitated and likely well mixed to enable uniform oxygen removal. Analysis data showing uniform oxygen reduction across the titanium sample was also added.
3. Replaced some figures and images to make them clearer, like Figures 4c and 4d.
4. Removed non-essential table columns and tightened language at various places for clarity and conciseness, as suggested.
5. Broadened the conclusions to highlight larger implications for titanium production as an alternative to the conventional Kroll process.
6. Additional data on the presence of yttrium oxyfluoride and its likely stability at high temperatures were included in Supplementary Information.

The authors have also provided reasonable justification for why precise quantification of yttrium and other rare earth contents in titanium is currently challenging. They are working on improving this. Overall, the authors have adequately addressed all major and minor revision suggestions by the reviewers. In my assessment, the revised manuscript reads well, has clarity in the technical details, and makes a compelling case for the new titanium deoxidation approach. The changes have improved it further.

Reviewer #2 (Remarks to the Author):

see attachments

Reviewer #3 (Remarks to the Author):

See the attached file.

Answers for comments by Reviewers

REVIEWER COMMENTS

Reviewer #2 (Remarks to the Author):

Review of T. H. Okabe et al.: „Direct Production of Low-Oxygen-Concentration Titanium from Molten Titanium”

General comment on authors:

The response of the authors to the reviewers' initial comments and suggestions and the corresponding changes in the manuscript are appreciated. In general, the revised manuscript is recommended for publication by reviewer #2. Nevertheless, reviewer #2 would like to provide a few **non-mandatory** suggestions which might further improve the manuscript. These suggestions are optional and their incorporation is up to the authors.

T. H. Okabe has published many important articles on alternative methods for titanium metal extraction during the last almost twenty years, e.g. on calciothermic and magnesiothermic reduction, usage of halides and electrochemical methods. In 2019 and 2020, respectively, he authored two review papers in this field and a book chapter (reference 1 in current work). Recently, he published several articles on the usage of rear earths in combination with halide fluxes, as kind of predecessors to the current work.

The co-authors have published several articles on extractive metallurgy of titanium as well. Therefore, it is safe to say that the authors of the current work have got a high expertise in this field.

General comments on the approach described in current work:

The key aspect of the approach is the deoxidation of titanium metal by rare earth metals (e.g. yttrium Y) and a halide flux (e.g. yttrium fluoride YF_3) with the formation of oxyhalides (e.g. YOF) instead of corresponding oxides, thus achieving significantly lower oxygen-contents.

The novelty of this approach is unquestionable.

The soundness of this approach is made plausible by suitable thermodynamic calculations and lab-scale experiments.

The paper is in principle suitable for publication in Nature Communications.

General critics/suggestions:

• Contamination of the deoxidized titanium by yttrium is significant, as evident from the current work and several preceding publications. This issue is not sufficiently addressed in the current work.

Experimentally obtained Y-contents are only roughly stated in the text, but instead they should be precisely given in Table 1 together with the contents of oxygen and nitrogen. In this way, the varying oxygen-contents become also more understandable (solubility product of Y and O, according to equation 3 and 4). According to section “post-experiment analysis”, Y and the other rare earth elements have been analyzed by ICP-AES and XRF, thus their exact contents should be given in Table 1. In addition, a general remark could be appropriate that the consequences of significant contents of the rare earths in titanium on its properties in the final application are not yet sufficiently investigated to date.

You have correctly pointed out that Table 1 should include the concentrations of Y, La, and Ce, alongside the amounts of oxygen and nitrogen. However, Y, La, and Ce have limited solubility in solid Ti, leading to significant segregation during cooling in the experiments. Determining the exact concentration of these rare-earth elements in molten Ti remains a challenge. We are working diligently to quantify the dissolved Y concentration accurately and are in the process of validating these findings, as illustrated in Figure R1.

In principal, despite the issue of strong segregation of Y etc. during solidification and cooling as mentioned by the authors, the overall contents of Y etc. in the metal should be measurable using sufficiently sized samples and appropriate methods like ICP-OES (as it has been done for oxygen and

nitrogen, probably by carrier gas hot extraction). These overall contents would correspond to the contents in the metallic melt. Fig. R1 already provides such overall Y-contents. However, reviewer #2 fully understands and accepts the authors' intention not to publish this confidential and undisclosed figure. The authors are encouraged to publish the contents of Y etc. in due time and appropriate context in the future.

Some experiments have yielded Ti with Y concentrations under 1000 mass ppm and oxygen levels also below 1000 mass ppm. This demonstrates that Ti can be effectively deoxidized in conditions that minimize Y contamination, a critical consideration for its industrial use.

Moreover, as you have also highlighted, the influence of rare-earth elements on the properties of Ti for end-use applications has not yet been fully investigated. The low-oxygen Ti produced by the current work may not be suitable for high-reliability applications such as aerospace-grade Ti alloys. Nevertheless, we are optimistic about the potential application of this Ti in consumer products, such as a material in devices like iPhones.

Fig. R1. Thermodynamic relationships between Y and oxygen concentrations in Ti–Y–O melt coexisting with solid Y_2O_3 phase or two phases of liquid YF_3 and solid $Y_4O_3F_6$ at 2000 K (1727 °C). (Unpublished data: Since it relates to intellectual property, such as know-how and experimental conditions, this figure should be confidential and undisclosed information at this time.)

Reviewer #2 fully understands and accepts the authors' intention not to publish any confidential and undisclosed information hence omitting Fig. R1. Nevertheless, it should be mentioned that this diagram is very beneficial for several reasons:

- a) The solubility product for Y-deoxidation provides two essential parameters of the process, namely the achievable contents of oxygen and yttrium.
- b) A rather good agreement of measurements and calculations is apparent.
- c) The advantage of much lower Y- and O-contents in the metal at the equilibrium Ti-Y-O-melt/ $Y_4O_3F_6$ / YF_3 is apparent compared to the equilibrium Ti-Y-O-melt/ Y_2O_3 as measured by Kobayashi 1998.

Reviewer #2 would like to encourage the authors to publish Fig. R1 in due time in the future.

*• **Temperature control in the experiments** is not clear. Induction melting, in particular with a cold crucible, generally is challenging regarding temperature control of the melt. The temperature in turn is crucial to the thermodynamic equilibrium of the deoxidation (e. g. equation 4). It should be stated how the temperature was measured (radiation pyrometer, thermocouple etc.) and which values or range were obtained.*

We did not use a radiation pyrometer or thermocouple for direct temperature measurement. However, the cold crucible induction melting furnace (induction skull melting furnace) used in this experiment is thought to have heated only to about 50°C above the melting point of the material. This is due to the large contact area between the water-cooled copper crucible and the molten Ti (350 g) and the presence of solid Ti at the interface. Previous measurements indicate that the furnace does not heat the metal to more than 100°C above its melting point. This is because the stirring of the melt and the resulting radiative heat loss are substantial relative to the power input.

The authors' reply on process temperature is comprehensive and convincing. Their method for estimation of temperature is sound, but probably not trivial to many readers. As the temperature is an essential parameter regarding both the process technique and the thermodynamic analysis, it is highly suggested to include this short text passage into the manuscript.

*• **Application of cold crucible ("skull") induction melting** possibly limits the potential for industrial scale-up and is very energy intensive. However, it is clear from the current state of the art and science that suitable refractory crucibles have not been found yet for high-Ti melts despite decades of intensive research and development. Y_2O_3 has been applied for Ti-melts in prior publications but would not be suitable in the current approach due to the need for saturation of the flux with Y_2O_3 in that case. So, it remains open what refractories could be used in the process sketched in Figure 6. Nevertheless, compared to conventional melting processes for titanium (vacuum arc remelting, electron melting and plasma arc melting), cold crucible induction melting could still be advantageous.*

In the current work, a cold crucible (induction skull melting) furnace was used, primarily to facilitate the rapid melting of the experimental sample, expedite the deoxidation reaction through the magnetic levitation and vigorous agitation of the Ti melt, and ensure the uniformity of the sample.

You are correct in pointing out that the application of cold crucible ("skull") induction melting does have its limitations for industrial scale-up. Currently, ISM furnaces capable of melting approximately 100 kg per batch are in practical use, and we have received information from KOBELCO Ltd. regarding a prototype furnace with the capacity to melt around 200 kg. Additionally, a method for semi-continuously extracting molten Ti from 20 kg batches to cast 100 kg ingots has been successfully developed and put into practice. However, as you note, for melting on the order of tons, plasma arc melting furnaces and electron beam melting furnaces would be more appropriate.

In scaling up and commercializing the process depicted in Figure 6, one viable approach might be to employ a plasma torch or a similar heating method that heats the Ti melt from above, especially when using a water-cooled copper crucible. Alternatively, using high-melting-point metals like Mo and W, which require external cooling, could also be considered. We are currently investigating the feasibility of using solid rare-earth fluorides or oxyfluorides as refractory linings in reaction vessels.

There are the following comments and suggestions:

- Equations (3): One should assume that the RE is dissolved in the Ti and that REX_3 and $REOX$ form a homogeneous mixture phase. At 1700°C Ti and Y are completely miscible. Then, the equation reads:

As a consequence, the reduced activities of the corresponding component have to be considered in the thermodynamic calculations.

- Figure 2b: E.g. according to the reaction

the mole fraction of x_O in Ti is given by

$$x_O = \frac{1}{\gamma_O} \left(\frac{K(T) a_{Y_2O_3}}{a_Y^2} \right)^{1/3}$$

with $K(T)$ equilibrium constant and γ_O the activity coefficient of O in Ti (which in principle also depends on x_O). The equation can also be written as

$$P_{O_2} = a_O = \left(\frac{K(T) a_{Y_2O_3}}{a_Y^2} \right)^{1/3}$$

(O_2 reference state for oxygen). So, the position of the hollow circles in Figure 2b depend on a_Y (and thereby on x_Y) and on $a_{Y_2O_3}$. Of course, the same is also the case for the other symbols. It is o.k. to make a reasonable choice for the activities (e.g. $a=1$) but it should be explained more clearly which one and why.

It is o.k. to make a reasonable choice for the activities (e.g. $a=1$) but it should be explained more clearly which one and why.

To assess the decrease in Y activity due to its dissolution in the Ti melt, calculation results with varying Y activities are shown alongside each other in Figs. 2a and 2c. To our knowledge, there are no reported measurements of the activities of Ti and Y in the Ti-Y binary system's liquid phase. Given that a substantial amount of YF_3 flux was added in the experiments, and the solubility of yttrium oxyfluoride in the liquid YF_3 phase is relatively low near the melting point of Ti, as cited in (Ref. 25) [Baek_2022], setting the activities of YF_3 and YOF to unity is considered reasonable. The analysis of Figure R1 suggests the absence of any significant discrepancy between our thermodynamic analysis and the experimental results.

We have added the following explanation after Line 152 in the thermodynamic calculation section of the manuscript:

From Line 152

➔ “...the current work. The activities of the deoxidant (Y), the flux (YF_3), and the reaction product (YOF)—with reference states corresponding to pure solid Y, pure liquid YF_3 , and pure solid YOF, respectively—were set to unity in principle. Calculations accounting for the decrease in Y activity due to its dissolution into the Ti melt are detailed in Figures 2a and 2c. Given the substantial amount of YF_3 flux used in the experiments and the relatively low solubility of yttrium oxyfluoride in the liquid YF_3 phase near Ti's melting point²⁵, it was deemed reasonable to assume unity for the activities of YF_3 and YOF.”

Minor comments and suggestions:

- *It's good practice to mention the reference state when using activities like in Figure 2a.*

The following explanation was added from L. 152 in the thermodynamic calculation section of the manuscript.

➔ “In the current work, the activities of the deoxidant (Y), the flux (YF_3), and the reaction product (YOF)—with reference states corresponding to pure solid Y, pure liquid YF_3 , and pure solid YOF, respectively—were set to unity in principle.”

• *Line 45 etc.: A further significant disadvantage of the conventional extraction route is the need for remelting the titanium sponge from Kroll process, thus increasing costs and energy demand significantly.*

Thank you for your comments on the revision and improvement of the text. We have added the above statement to the text at Line 43–45 as follows:

➔ A further significant disadvantage of the conventional extraction route is the need to melt the Ti sponge produced from the Kroll process, which significantly increases costs and energy consumption.

• *Line 48: Expression “...both of which exhibit higher chemical affinities with oxygen than Ti” is imprecise and not quite right. As also evident from Figs. 2b and 2c, due to the significant oxygen solubility of titanium, the oxygen potential pO_2 required for reduction/deoxidation is delicately determined by the pursued oxygen content.*

You are correct. The text has been revised as follows. “...both of which exhibit higher chemical affinities with oxygen than Ti” on Lines 50–51 has been replaced with “...both of which exhibit strong chemical affinities with oxygen.”

• *Line 60 etc.: Regarding aluminothermy, the significant contamination of titanium with aluminium could be mentioned, according to the solubility product in the system Ti-Al-O. Furthermore, the seminal work of Maeda et al. (reference 22) could also be cited here.*

As you suggested, we have included a reference to Prof. Maeda's work in our discussion of ATR-Ti production (Line 52) and have added the following text after Line 62:

➔ “Furthermore, ATR-Ti is heavily contaminated with Al, rendering it unsuitable for commercial applications in its current state⁷.”

• *Line 60 etc.: Why is calciothermy not mentioned beside aluminothermy and magnesiothermy? Due to costs for calcium?*

Yes, we did not include calcium (Ca) in our discussion due to its prohibitively high production costs. In the past, we have conducted research on the reduction of TiO_2 using metallic Ca and developed a process that could produce metallic Ti with a low oxygen concentration of around 500 mass ppm, as well as minimal Ca contamination. However, we found that using Ca as a reductant is

not economically feasible since a technology for achieving the low-cost production of metallic Ca has yet to be established.

• Line 183: "... while suppressing the incorporation of Y into the deoxidized Ti melt". It would be better to write "... while minimising the incorporation of Y into the deoxidized Ti melt". A "suppressing" of Y-pick up is thermodynamically impossible.

Thank you for pointing this out. The manuscript has been updated on Line 208 make the suggested revision.

• Line 200 (Fig. 3): Although the apparatus dimensions are later given in the section "experimental setup", why not draw the internal diameter of the crucible or a scale right in the sketch?

Thank you for pointing this out. We will update the manuscript to make the suggested revision.

• Line 210 (Fig. 4c): Is the outer "ring" of the sample the solidified flux? (could be mentioned in the corresponding caption)

As you have intuited, the outer "ring" in the image is indeed solidified flux. Acknowledging your observation that Fig. 4c is not very clear, we have replaced it with a new photo that clearly marks the metallic Ti and the flux.

Fig. 4c.

• *Figure 4d: The Figure is difficult to understand.*

As you have pointed out, Fig. 4d is indeed difficult to understand, so we have replaced it with the following photo. The oxygen analysis results shown on the left in the photo (Fig. R2) are provided for reference only.

Fig. 4d.

Fig. R2. Distribution of oxygen and nitrogen concentration in the Ti sample of Exp. #1. Oxygen and nitrogen concentrations were analyzed at 10 points across 6 locations, indicated by black dots. The results are summarized in the table. (The initial oxygen concentration in the Ti sample was approximately 1000 mass ppm.) Across all analysis points, including the center and near the bottom of the sample, the oxygen concentration consistently decreased to around the 200 mass ppm level. This uniform reduction throughout suggests that the deoxidation reaction occurred evenly across the entire sample.

• Line 218 (Table 1): The last column (“Experimental code”) contains no information relevant to the public, unless it refers to supplementary data additionally provided. It would be very useful to give also the Y-, La-, Ce-content in the final material.

You are correct in noting that the “Experimental code” column, while important for our experimental management, holds no significance for the readers. Accordingly, it has been removed.

Regarding your suggestion to include concentrations of Y, La, and Ce in Table 1 along with the oxygen and nitrogen contents, we agree with you in principle. However, unlike oxygen, Y, La, and Ce are only minimally soluble in solid Ti, leading to significant segregation during the cooling phase of the experiments. Currently, we are working to quantify the concentrations of these rare-earth elements dissolved in the Ti melt in more precise terms and to assess their reliability, as illustrated in Figure R1. In some experiments, we obtained Ti samples with Y concentrations of less than 1000 mass ppm and oxygen concentrations also below 1000 mass ppm. Demonstrating that Ti can be effectively deoxidized under conditions that minimize Y contamination is a key insight for the industrial application of deoxidized Ti.

• Line 260 (Fig. 6): Designation “Flux (CaF₂, etc.)” could be extended by “YF₃”, according to the approach and findings of the current work. What are the designated refractory materials?

Thank you for your suggestion. We have accordingly made the following revision in Fig. 6:

➔ Flux (CaF₂, etc.) ➔ Flux (YF₃, CaF₂, etc.)

In scaling up and commercializing the process depicted in Figure 6, one viable approach might be to employ a plasma torch or a similar heating method that heats the Ti melt from above, especially when using a water-cooled copper crucible. Alternatively, using high-melting-point metals like Mo

and W, which require external cooling, could also be considered. We are currently investigating the feasibility of using solid rare-earth fluorides or oxyfluorides as refractory linings in reaction vessels.

- *Line 296: Pressure of argon atmosphere (e.g., “ambient”)?*

Thank you for your observation. The experiments were conducted in an Ar atmosphere at a pressure of 500 Torr. We will update the manuscript to make the suggested revision. See Line 332.

- *Line 356: “...seoxidation” (typo).*

Thank you for pointing this out. We will update the manuscript to make the suggested revision. See Line 394.

Thank you very much for your highly constructive and appropriate guidance and suggestions. Your comments will be invaluable not only for the current work but also for our future research endeavors. We are deeply grateful.

The following are items that were not directed by the reviewers, but the items that the authors felt the need to revise upon reviewing.

Line 232

agitated well by pinch force with induction heating,

=>

agitated well by Lorenz force (pinch force) with induction heating,

Line 234

subjected to the pinch force. => subjected to the Lorenz force.

Line 245

due to induction melting. => after induction melting.

Line 426

Mr. Yoichi Takashina of Toho Technical => Mr. Yoichi Warashina of Toho Technical

The method used for the levitation and agitation of the Ti melt during the melting of Ti samples in the current work is an important technique. The slit in the water-cooled copper crucible, which facilitates the introduction of high-frequency waves as depicted in the original Fig. 4a of the apparatus, is currently considered confidential and undisclosed information due to its association with intellectual property, such as experimental know-how and conditions. Consequently, in the revised Fig. 4a, the photograph of the slit in the water-cooled copper crucible has been modified with blurring for image confidentiality.

REDACTED

Photographs of a Ti sample after melting and deoxidation using a cold crucible induction melting furnace (Exp. #1 in Table 1). a, The sample in the cold crucible after melting; b, the side and c, top views; and d, Sectioned Ti sample pieces for analysis. The metal sample was separated from the flux phase after induction melting.

Reviewer #2 fully understands and accepts the authors' intention not to publish any confidential and undisclosed information related to intellectual property, hence blurring Fig. 4a. However, editing the image has several significant disadvantages:

- a) Information on the slits is also available implicitly from Figs. 3 and 4b anyway.
- b) A cold crucible without slits just would not work physically, hence the reader would be puzzled by Fig. 4a.
- c) Without the slits, Fig. 4a would be inconsistent with the scheme in Fig. 3, hence the reader would be puzzled.
- d) In general, if a photograph is edited retroactively in a significant way, this should be stated.

It is suggested to fully remove Fig. 4a because of the abovementioned issues as well as the fact that it doesn't provide significant additional information over the other related figures, in particular over the scheme in Fig. 3.

*** See Nature Portfolio's author and referees' website at www.nature.com/authors <<http://www.nature.com/authors>> for information about policies, services and author benefits.*

This email has been sent through the Springer Nature Tracking System NY-610A-NPG&MTS.

Confidentiality Statement:

This e-mail is confidential and subject to copyright. Any unauthorized use or disclosure of its contents is prohibited. If you have received this email in error please notify our Manuscript Tracking System Helpdesk team at <http://platformsupport.nature.com>

Details of the confidentiality and pre-publicity policy may be found here <http://www.nature.com/authors/policies/confidentiality.html>

Privacy Policy <<http://www.nature.com/info/privacy.html>> | Update Profile <<https://mts-ncomms.nature.com>>

DISCLAIMER: This e-mail is confidential and should not be used by anyone who is not the original intended recipient. If you have received this e-mail in error please inform the sender and delete it from your mailbox or any other storage mechanism. Springer Nature America, Inc. does not accept liability for any statements made which are clearly the sender's own and not expressly made on behalf of Springer Nature America, Inc. or one of their agents.

Please note that neither Springer Nature America, Inc. or any of its agents accept any responsibility for viruses that may be contained in this e-mail or its attachments and it is your responsibility to scan the e-mail and attachments (if any).

Reviewer #3 (Remarks to the Author):

This paper presents a method and experimental examples of the deoxidation of molten titanium.

Merits:

- 1. Deoxidation is a very challenging problem for Ti. The low oxygen content, <200 ppm, presented in the article is remarkable.*
- 2. The process presented could be used for the production of primary Ti as well as the deoxidation of scrap Ti.*

Detractors:

- 1. The novelty of the presented work is questionable. The authors' group has been very active in this area. I easily found, using Google search, over ten publications by the group on the general subject of using rare earth elements, specifically Y, to deoxidate Ti. Some are for deoxidate solid Ti, some molten Ti. I tried to see if the use of flux to form Y oxychloride is novel, but, the following publication seems to have significant similarity.*

TY - JOUR

AU - Okabe, Toru

AU - Taninouchi, Yu-ki

AU - Zheng, Chenyi

PY - 2018/08/31

SP -

TI - Thermodynamic Analysis of Deoxidation of Titanium Through the Formation of Rare-Earth Oxyfluorides

VL - 49 DO - 10.1007/s11663-018-1386-5 JO - Metallurgical and Materials Transactions B

As you have pointed out, we have conducted a series of studies over the years focused on removing oxygen from *solid* Ti. These studies primarily involved extracting oxygen from **the surface of solid Ti** at temperatures around 1300 K (1027 °C) by utilizing diffusion within the solid Ti. It has been demonstrated that Ti with oxygen concentrations **below 100 mass ppm** can be produced at around 1300 K (1027 °C).

In contrast, **the current work introduces** a novel method for directly removing oxygen **from liquid Ti at high temperatures above its melting point**. The deoxidation of liquid Ti has proven both thermodynamically and technically challenging, and the general understanding until now has been that producing Ti with oxygen levels below 1000 mass ppm by deoxidating Ti in a molten state was not feasible, as indicated in Ref. 9 [Kobayashi_1998] and Ref. 10 [Fukuzawa_1999], among others.

We believe the new technique developed in the current work, which is capable of directly producing Ti with low oxygen concentrations of around 200 mass ppm from the melt, represents a significant breakthrough.

The reviewer has responded to this comment above appropriately. Yes, the previous publication focused on solid state deoxidation, and the current manuscript focused on the deoxidation of liquid Ti. However, it still makes it not so much a brand-new breakthrough, but rather, an extension of the method used for the solid into the liquid state of the same metal. There is no brand-new method of concept.

However, this manuscript is still a nice work. I would support the publication if the editor chose to do so.

Additionally, in my search for the background on the deoxidation of Ti, I found on the internet that IperionX of Australia has been promoting a technology using Mg and H₂ to do the same as this manuscript claims to do. The authors seemed to have ignored it. It is curious why the authors left it out of their background descriptions.

- 2. The presentation of the article is a little unconventional. The authors presented examples number 1, 2, 3, etc. It reads like a patent application. The style makes it difficult to grasp the method's principles beyond the data report.*

In response to your comment, we have revised the table, changing “No.” to “Exp. #” to ensure consistency with the experimental descriptions elsewhere in the text (e.g., Exp. #1, Exp. #2). Additionally, based on feedback from another reviewer, we have removed the “Experimental code” from the table.

OK. It is not my style to write a scientific manuscript this way, but I will let it pass.

3. Another very minor point. XRD cannot be used to determine if residual oxides of other elements are present. It is not sensitive enough.

In response to your point about the limitations of XRD, we noted that the XRD measurement of the flux displayed distinct diffraction peaks for yttrium oxyfluoride that differed from those of Y and YF_3 . This led us to conclude that YOF and other oxyfluorides formed after the deoxidation experiments. While it is true, as you mentioned, that XRD cannot provide a quantitative assessment of Y_2O_3 , we observed no diffraction peaks of Y_2O_3 in the flux. Additionally, SEM analysis of the fluxes confirmed the presence of oxyfluorides but did not detect Y_2O_3 .

To clarify this point further, we have added enlarged views of the XRD patterns in Fig. 5b to show the absence of Y_2O_3 in the flux. We have also included the SEM-EDS results of the fluxes in Supplementary Figure 6 and Supplementary Table 5.

I don't think the authors appreciated the point that neither XRD nor EDS is sensitive enough for trace element analysis. There will be residual Y in the material after using their method. More sensitive techniques such as IPS must be used to measure the trace amount of such elements accurately. However, again, I will let it pass.

Fig. 5 XRD patterns of YF_3 fluxes before and after deoxidation. The intensities are shown in (a) wide and (b) narrow diffraction angle scales. Various yttrium oxyfluoride phases were detected in the fluxes after the deoxidation experiments, whereas no yttrium and titanium oxide phases were observed, which shows that the $\text{Y}/\text{YOF}/\text{YF}_3$ equilibrium was established in the current systems.

Answers for comments by Reviewers

Comments in **blue texts** are the first comments for the submitted manuscript from reviewers.

Comments in **red texts** are the second comments from reviewers on the revised manuscript.

Texts in black are the reply from the author to the (first and) second comments.

REVIEWER COMMENTS

Reviewer #1 (Remarks to the Author):

Based on my review, the authors have thoroughly addressed the reviewers' comments and suggestions to improve the manuscript. Here are some of the key changes made:

- 1. More details on the thermodynamic calculations, data sources for Y-O-F system properties at high temperatures, and assumptions on activity coefficients were added. This strengthens the analysis.*
- 2. Clarified that the titanium melt was strongly agitated and likely well mixed to enable uniform oxygen removal. Analysis data showing uniform oxygen reduction across the titanium sample was also added.*
- 3. Replaced some figures and images to make them clearer, like Figures 4c and 4d.*
- 4. Removed non-essential table columns and tightened language at various places for clarity and conciseness, as suggested.*
- 5. Broadened the conclusions to highlight larger implications for titanium production as an alternative to the conventional Kroll process.*
- 6. Additional data on the presence of yttrium oxyfluoride and its likely stability at high temperatures were included in Supplementary Information.*

The authors have also provided reasonable justification for why precise quantification of yttrium and other rare earth contents in titanium is currently challenging. They are working on improving this. Overall, the authors have adequately addressed all major and minor revision suggestions by the reviewers. In my assessment, the revised manuscript reads well, has clarity in the technical details, and makes a compelling case for the new titanium deoxidation approach. The changes have improved it further.

Answer for the second comment

Thank you very much for your highly constructive and appropriate guidance and suggestions. Your comments will be invaluable not only for the current work but also for our future research endeavors. We deeply appreciate your valuable comments and suggestions.

Reviewer #2 (Remarks to the Author):

Review of T. H. Okabe et al.: „Direct Production of Low-Oxygen-Concentration Titanium from Molten Titanium”

General comment on authors:

The response of the authors to the reviewers' initial comments and suggestions and the corresponding changes in the manuscript are appreciated. In general, the revised manuscript is recommended for publication by reviewer #2. Nevertheless, reviewer #2 would like to provide a few **non-mandatory** suggestions which might further improve the manuscript. These suggestions are optional and their incorporation is up to the authors.

T. H. Okabe has published many important articles on alternative methods for titanium metal extraction during the last almost twenty years, e.g. on calciothermic and magnesiothermic reduction, usage of halides and electrochemical methods. In 2019 and 2020, respectively, he authored two review papers in this field and a book chapter (reference 1 in current work). Recently, he published several articles on the usage of rare earths in combination with halide fluxes, as kind of predecessors to the current work.

The co-authors have published several articles on extractive metallurgy of titanium as well. Therefore, it is safe to say that the authors of the current work have got a high expertise in this field. General comments on the approach described in current work:

The key aspect of the approach is the deoxidation of titanium metal by rare earth metals (e.g. yttrium Y) and a halide flux (e.g. yttrium fluoride YF₃) with the formation of oxyhalides (e. g. YOF) instead of corresponding oxides, thus achieving significantly lower oxygen-contents.

The novelty of this approach is unquestionable.

The soundness of this approach is made plausible by suitable thermodynamic calculations and lab-scale experiments.

The paper is in principle suitable for publication in Nature Communications.

General critics/suggestions:

• **Contamination of the deoxidized titanium by yttrium** is significant, as evident from the current work and several preceding publications. This issue is not sufficiently addressed in the current work.

Experimentally obtained Y-contents are only roughly stated in the text, but instead they should be precisely given in Table 1 together with the contents of oxygen and nitrogen. In this way, the varying oxygen-contents become also more understandable (solubility product of Y and O, according to equation 3 and 4). According to section “post-experiment analysis”, Y and the other rare earth elements have been analyzed by ICP-AES and XRF, thus their exact contents should be given in Table 1. In addition, a general remark could be appropriate that the consequences of significant contents of the rare earths in titanium on its properties in the final application are not yet sufficiently investigated to date.

You have correctly pointed out that Table 1 should include the concentrations of Y, La, and Ce, alongside the amounts of oxygen and nitrogen. However, Y, La, and Ce have limited solubility in solid Ti, leading to significant segregation during cooling in the experiments. Determining the exact concentration of these rare-earth elements in molten Ti remains a challenge. We are working diligently to quantify the dissolved Y concentration accurately and are in the process of validating these findings, as illustrated in Figure R1.

In principal, despite the issue of strong segregation of Y etc. during solidification and cooling as mentioned by the authors, the overall contents of Y etc. in the metal should be measurable using sufficiently sized samples and appropriate methods like ICP-OES (as it has been done for oxygen and nitrogen, probably by carrier gas hot extraction). These overall contents would correspond to the contents in the metallic melt. Fig. R1 already provides such overall Y-contents. However, reviewer #2 fully understands and accepts the authors' intention not to publish this confidential and undisclosed figure. The authors are encouraged to publish the contents of Y etc. in due time and appropriate context

in the future.

Answer for the second comment

Y is soluble in liquid Ti, but very little Y is soluble in solid Ti, resulting in very large segregation during solidification and Y precipitation at the grain boundaries of Ti crystals. Therefore, the Y concentration in the solidified Ti varies greatly depending on the cooling conditions and locations, and the analysis of the Y concentration in the ingot obtained after melting and cooling has a large error. In the future, it will be necessary to sample a small analytical specimen directly from Ti-Y melts in order to determine the relationship between Y and oxygen concentrations in those liquid samples.

Thank you for understanding and accepting our situation and intention not to publish this confidential and undisclosed figure. In the future, the authors will publish the relationship between the contents of oxygen and Y in the Ti melt when the situation allows us to disclose our know-how and IPs.

Some experiments have yielded Ti with Y concentrations under 1000 mass ppm and oxygen levels also below 1000 mass ppm. This demonstrates that Ti can be effectively deoxidized in conditions that minimize Y contamination, a critical consideration for its industrial use.

Moreover, as you have also highlighted, the influence of rare-earth elements on the properties of Ti for end-use applications has not yet been fully investigated. The low-oxygen Ti produced by the current work may not be suitable for high-reliability applications such as aerospace-grade Ti alloys. Nevertheless, we are optimistic about the potential application of this Ti in consumer products, such as a material in devices like iPhones.

Fig. R1. Thermodynamic relationships between Y and oxygen concentrations in Ti–Y–O melt coexisting with solid Y_2O_3 phase or two phases of liquid YF_3 and solid $Y_4O_3F_6$ at 2000 K (1727 °C). (Unpublished data: Since it relates to intellectual property, such as know-how and experimental conditions, this figure should be confidential and undisclosed information at this time.)

Reviewer #2 fully understands and accepts the authors' intention not to publish any confidential and undisclosed information hence omitting Fig. R1. Nevertheless, it should be mentioned that this diagram is very beneficial for several reasons:

- a) The solubility product for Y-deoxidation provides two essential parameters of the process, namely the achievable contents of oxygen and yttrium.
- b) A rather good agreement of measurements and calculations is apparent.
- c) The advantage of much lower Y- and O-contents in the metal at the equilibrium Ti-Y-O-melt/ $Y_4O_3F_6/YF_3$ is apparent compared to the equilibrium Ti-Y-O-melt/ Y_2O_3 as measured by Kobayashi 1998.

Reviewer #2 would like to encourage the authors to publish Fig. R1 in due time in the future.

Answer for the second comment

Thank you very much for your valuable comments and suggestions. As you pointed out, the advantage of much lower Y- and O-contents in the metal at the equilibrium Ti-Y-O melt / $Y_4O_3F_6$ / YF_3 is apparent compared to the equilibrium Ti-Y-O melt / Y_2O_3 as measured by Kobayashi 1998.

Thank you for encouraging us to publish Fig. R1 in due time in the future. We will do publish, however, for the time being, we need to overcome sampling problem of Ti-Y melt for precise determination of Y in Ti.

• Temperature control in the experiments is not clear. Induction melting, in particular with a cold crucible, generally is challenging regarding temperature control of the melt. The temperature in turn is crucial to the thermodynamic equilibrium of the deoxidation (e. g. equation 4). It should be stated how the temperature was measured (radiation pyrometer, thermocouple etc.) and which values or range were obtained.

We did not use a radiation pyrometer or thermocouple for direct temperature measurement. However, the cold crucible induction melting furnace (induction skull melting furnace) used in this experiment is thought to have heated only to about 50°C above the melting point of the material. This is due to the large contact area between the water-cooled copper crucible and the molten Ti (350 g) and the presence of solid Ti at the interface. Previous measurements indicate that the furnace does not heat the metal to more than 100°C above its melting point. This is because the stirring of the melt and the resulting radiative heat loss are substantial relative to the power input.

The authors' reply on process temperature is comprehensive and convincing. Their method for estimation of temperature is sound, but probably not trivial to many readers. As the temperature is an essential parameter regarding both the process technique and the thermodynamic analysis, it is highly suggested to include this short text passage into the manuscript.

Answer for the second comment

Based on your comments, following text are added to the manuscript Line 328.

The cold crucible induction melting furnace (induction skull melting furnace) used in the experiment is thought to have heated only to about 50 K above the melting point of the material. This is due to the large contact area between the water-cooled Cu crucible and the molten Ti (350 g) and the presence of solid material at the interface. Previous measurements indicate that the furnace does not heat the metal to more than 100 K above its melting point. This is because the stirring of the melt and the resulting radiative heat loss are substantial relative to the power input.

The method used for the levitation and agitation of the Ti melt during the melting of Ti samples in the current work is an important technique. The slit in the water-cooled copper crucible, which facilitates the introduction of high-frequency waves as depicted in the original Fig. 4a of the apparatus, is currently considered confidential and undisclosed information due to its association with intellectual property, such as experimental know-how and conditions. Consequently, in the revised Fig. 4a, the photograph of the slit in the water-cooled copper crucible has been modified with blurring for image confidentiality.

Fig. 4

Photographs of a Ti sample after melting and deoxidation using a cold crucible induction melting furnace (Exp. #1 in Table 1). A, The sample in the cold crucible after melting; b, the side and c, top views; and d, Sectioned Ti sample pieces for analysis. The metal sample was separated from the flux phase after induction melting.

Reviewer #2 fully understands and accepts the authors' intention not to publish any confidential and undisclosed information related to intellectual property, hence blurring Fig. 4a. However, editing the image has several significant disadvantages:

- a) Information on the slits is also available implicitly from Figs. 3 and 4b anyway.
- b) A cold crucible without slits just would not work physically, hence the reader would be puzzled by Fig. 4a.
- c) Without the slits, Fig. 4a would be inconsistent with the scheme in Fig. 3, hence the reader would be puzzled.
- d) In general, if a photograph is edited retroactively in a significant way, this should be stated.

It is suggested to fully remove Fig. 4a because of the abovementioned issues as well as the fact that it doesn't provide significant additional information over the other related figures, in particular over the scheme in Fig. 3.

Answer for the second comment

Based on your comments, we realized it is not good to modify the photo. As you pointed out, the modified photo with blurring (the image without the slits in the water-cooled Cu crucible) is puzzling to the reader. Considering the IP issue, we decided to use non-modified image with restricted image area. The following photo is used for the manuscript. Thank you again for your valuable comments.

REDACTED

Fig. 4

Photographs of a Ti sample after melting and deoxidation using a cold crucible induction melting furnace (Exp. #1 in Table 1). a, The sample in the cold crucible after melting; b, the side and c, top views; and d, Sectioned Ti sample pieces for analysis. The metal sample was separated from the flux phase after induction melting.

Supplementary Figure 2

Analysis positions of the Ti sample in Exp. #1. The Ti sample was analyzed for oxygen and nitrogen concentrations by **LECO**. As shown in Supplementary Table 2, the deoxidation reaction proceeded uniformly in the entire sample. The red dotted line surrounds the Ti sample after deoxidation, which was cut on a base plate and divided into small pieces.

“LECO” designates a specific company, whereas in the current context, the general method of chemical analysis is more relevant. Probably the method of carrier gas hot extraction (CGHE) has been applied? It is suggested to state this instead of “LECO” in Figs. 2, 3, and 4 (within the figures as well as in the corresponding captions). If the authors would like to provide in full detail, they could state “... by carrier gas hot extraction (LECO + Type of machine used)...”

Answer for the second comment

Based on your comments, we modified the description of the analytical method of carrier gas hot extraction (CGHE) instead of “LECO.” We also removed the word “LECO” from the photos in the Supplemental Information.

LECO ==> the inert gas fusion method (LECO ON836).

Original photo

Modified photo

Supplementary Figure 2

Analysis positions of the Ti sample in Exp. #1. The Ti sample was analyzed for oxygen and nitrogen concentrations by **the inert gas fusion method (LECO ON836)**. As shown in Supplementary Table 2, the deoxidation reaction proceeded uniformly in the entire sample. The red dotted line surrounds the Ti sample after deoxidation, which was cut on a base plate and divided into small pieces.

Supplementary Figure 5

XRD patterns of fluxes after the experiments and initial YF3 reagent. Diffraction angles (2θ): (a) 10–90 degrees, (b) 27–~~30~~29 degrees, (c) 30–35 degrees, and (d) 45–50 degrees. The YF_3 peaks seemed to have shifted to the higher angle side. In the Exp. #8, a new large peak was observed at high angle near 28 degrees. It was considered to be a peak of a compound consisting of Y–O–F. Peaks around 32.5 degrees and 33 degrees in the fluxes of Exps. #1 and #8 were thought to be due to a compound composed of Y–O–F. There was a peak around 46.5 degree in the flux of Exp. #8, which may be attributed to a compound composed of Y–O–F. Y_2O_3 peaks were not observed at 29.4 degrees, 34 degrees, and 48.9 degrees.

Typo in this caption? (see highlighted text)

Answer for the second comment

As you pointed out, “30” degrees is correct.

Thank you very much for your highly constructive and appropriate guidance and suggestions. Your comments will be invaluable not only for the current work but also for our future research endeavors. We deeply appreciate your valuable comments and suggestions.

Reviewer #3 (Remarks to the Author):

This paper presents a method and experimental examples of the deoxidation of molten titanium.

Merits:

- 1. Deoxidation is a very challenging problem for Ti. The low oxygen content, <200 ppm, presented in the article is remarkable.*
- 2. The process presented could be used for the production of primary Ti as well as the deoxidation of scrap Ti.*

Detractors:

- 1. The novelty of the presented work is questionable. The authors' group has been very active in this area. I easily found, using Google search, over ten publications by the group on the general subject of using rare earth elements, specifically Y, to deoxidate Ti. Some are for deoxidate solid Ti, some molten Ti. I tried to see if the use of flux to form Y oxychloride is novel, but, the following publication seems to have significant similarity.*

TY - JOUR

AU - Okabe, Toru

AU - Taninouchi, Yu-ki

AU - Zheng, Chenyi

PY - 2018/08/31

SP -

TI - Thermodynamic Analysis of Deoxidation of Titanium Through the Formation of Rare-Earth Oxyfluorides

VL - 49 DO - 10.1007/s11663-018-1386-5 JO - Metallurgical and Materials Transactions B

As you have pointed out, we have conducted a series of studies over the years focused on removing oxygen from solid Ti. These studies primarily involved extracting oxygen from the surface of solid Ti at temperatures around 1300 K (1027 °C) by utilizing diffusion within the solid Ti. It has been demonstrated that Ti with oxygen concentrations below 100 mass ppm can be produced at around 1300 K (1027 °C).

In contrast, the current work introduces a novel method for directly removing oxygen from liquid Ti at high temperatures above its melting point. The deoxidation of liquid Ti has proven both thermodynamically and technically challenging, and the general understanding until now has been that producing Ti with oxygen levels below 1000 mass ppm by deoxidating Ti in a molten state was not feasible, as indicated in Ref. 9 [Kobayashi_1998] and Ref. 10 [Fukuzawa_1999], among others.

We believe the new technique developed in the current work, which is capable of directly producing Ti with low oxygen concentrations of around 200 mass ppm from the melt, represents a significant breakthrough.

The reviewer has responded to this comment above appropriately. Yes, the previous publication focused on solid state deoxidation, and the current manuscript focused on the deoxidation of liquid Ti. However, it still makes it not so much a brand-new breakthrough, but rather, an extension of the method used for the solid into the liquid state of the same metal. There is no brand-new method of concept.

However, this manuscript is still a nice work. I would support the publication if the editor chose to do so.

Additionally, in my search for the background on the deoxidation of Ti, I found on the internet that IperionX of Australia has been promoting a technology using Mg and H₂ to do the same as this manuscript claims to do. The authors seemed to have ignored it. It is curious why the authors left it out of their background descriptions.

Answer for the second comment

IperionX's deoxidation technology from solid Ti powders, originally developed by Prof. Z. Fang of the University of Utah, is a method of introducing hydrogen (H₂) gas into solid Ti to increase the activity coefficient of oxygen in solid Ti, and remove oxygen directly from Ti using magnesium (Mg) (which has low deoxidizing ability). This method, also known as the hydrogen assisted magnesiothermic reduction (HAMR) method, can deoxidize solid Ti down to 500 mass ppm O. It has great commercial value because it utilized inexpensive H₂ and Mg. However, due to Mg's high vapor pressure and low deoxidizing ability, it is in principle impossible to deoxidize "liquid" Ti at high temperature. A vast amount of research has been carried out in the past on methods for removing oxygen from solid Ti, and it has been demonstrated that deoxidation is possible up to an oxygen concentration of about 500 mass ppm, even when using metallic calcium (Ca), for example. In this context, the past research works on the removal of oxygen from solid Ti are limited by citing following comprehensive reference.

1. in Fang, Z. Z., Froes, F. H. & Zhang, Y. Extractive Metallurgy of Titanium (Elsevier, 2020).

On the other hand, the method used in this study, which utilizes the formation reaction of oxyhalides, was found for the first time to be able to deoxidize liquid Ti to about 200 mass ppm O, as demonstrated in this study. When this method is applied to the deoxidation of solid Ti, it can deoxidize down to a level of about 20 mass ppm O.

2. The presentation of the article is a little unconventional. The authors presented examples number 1, 2, 3, etc. It reads like a patent application. The style makes it difficult to grasp the method's principles beyond the data report.

In response to your comment, we have revised the table, changing "No." to "Exp. #" to ensure consistency with the experimental descriptions elsewhere in the text (e.g., Exp. #1, Exp. #2). Additionally, based on feedback from another reviewer, we have removed the "Experimental code" from the table.

OK. It is not my style to write a scientific manuscript this way, but I will let it pass.

Answer for the second comment

Thank you for your understanding.

3. Another very minor point. XRD cannot be used to determine if residual oxides of other elements are present. It is not sensitive enough.

In response to your point about the limitations of XRD, we noted that the XRD measurement of the flux displayed distinct diffraction peaks for yttrium oxyfluoride that differed from those of Y and YF₃. This led us to conclude that YOF and other oxyfluorides formed after the deoxidation

experiments. While it is true, as you mentioned, that XRD cannot provide a quantitative assessment of Y_2O_3 , we observed no diffraction peaks of Y_2O_3 in the flux. Additionally, SEM analysis of the fluxes confirmed the presence of oxyfluorides but did not detect Y_2O_3 .

To clarify this point further, we have added enlarged views of the XRD patterns in Fig. 5b to show the absence of Y_2O_3 in the flux. We have also included the SEM-EDS results of the fluxes in Supplementary Figure 6 and Supplementary Table 5.

I don't think the authors appreciated the point that neither XRD nor EDS is sensitive enough for trace element analysis. There will be residual Y in the material after using their method. More sensitive techniques such as IPS must be used to measure the trace amount of such elements accurately. However, again, I will let it pass.

Answer for the second comment

We understand that neither XRD nor EDS are sensitive enough for trace element analysis. However, in this study, it is important to demonstrate that the deoxidation reaction of Ti proceeded under YF_3/YOF equilibrium. For this reason, analysis by XRD was used to determine the major phases of the reaction products.

Thank you very much for your highly constructive and appropriate guidance and suggestions. Your comments will be invaluable not only for the current work but also for our future research endeavors. We are deeply grateful.

The following are items that were not directed by the reviewers, but the items that the authors felt the need to revise upon reviewing.

Line 213

In the melting experiments (Exps. #9, #10, #11, and #12),

==>

In the melting experiments (Exps. #9, #10, and #11),

Please refrain from using words such as new/novel/first, when referring to the scientific findings. Please also remove exaggerated language such as 'extremely'/'outstanding'. Please also remove or replace the following.

- word 'innovative' in line 15

==> The word innovative is considered appropriate in the context

- word 'uniquely' in line 22

==> The word "uniquely" deleted from the text.

- word 'innovation' in line 25

==> The word "innovation" was replaced by "technology"

- word 'novel' in line 96

==> The word "novel" deleted from the text, and text modified as follows

a method of effectively reducing impurity oxygen levels in molten Ti below 1000 mass ppm (0.1 mass%), which does not exist in the past,

- word 'new' in line 297

==> The word "new" deleted from the text.

Remaining reviewer comments

Our guidance:

We advise you to respond to the remaining reviewers' concerns (attached). In particular, we ask you to address the concern of using 'LECO' and retroactively edited Fig. 4 raised by reviewer #2. If you need to keep Fig. 4 in the main text or Supplementary Information, please clearly explain what has been removed retroactively from the image and the reasoning behind the edit (e.g. intellectual property). We also suggest sharing Fig. R1 given the reviewers' suggestions even in part, if allowed.

Based on the reviewer and editor's comments, we modified the description of the analytical method of carrier gas hot extraction (CGHE) has instead of "LECO". We also removed the word "LECO" from the photos in the Supplemental Information.

LECO ==> the inert gas fusion method (LECO ON836).